# MIN-MAX OPTIMIZATION WITHOUT GRADIENTS: CONVERGENCE AND APPLICATIONS TO ADVERSARIAL ML

## ABSTRACT

In this paper, we study the problem of constrained robust (min-max) optimization in a black-box setting, where the desired optimizer cannot access the gradients of the objective function but may query its values. We present a principled optimization framework, integrating a zeroth-order (ZO) gradient estimator with an alternating projected stochastic gradient descent-ascent method, where the former only requires a small number of function queries and the later needs just one-step descent/ascent update. We show that the proposed framework, referred to as ZO-Min-Max, has a sub-linear convergence rate under mild conditions and scales gracefully with problem size. From an application side, we explore a promising connection between black-box min-max optimization and black-box evasion and poisoning attacks in adversarial machine learning (ML). Our empirical evaluations on these use cases demonstrate the effectiveness of our approach and its scalability to dimensions that prohibit using recent black-box solvers.

## 1 INTRODUCTION

In numerous real-world applications, one is faced with various forms of adversary that are not accounted for by *standard* optimization algorithms. For instance, when training a machine learning model on user-provided data, malicious users can carry out a data poisoning attack: providing false data with the aim of corrupting the learned model (Steinhardt et al., 2017; Tran et al., 2018; Jagielski et al., 2018). At inference time, malicious users can evade detection of multiple models in the form of adversarial example attacks (Goodfellow et al., 2014; Liu et al., 2016; 2018a). *Min-max (robust)* optimization is a natural framework to address adversarial (worst-case) robustness (Madry et al., 2017b; Al-Dujaili et al., 2018b). It converts a standard minimization problem into a composition of an inner maximization problem and an outer minimization problem.

Min-max optimization problems have been studied for multiple decades (Wald, 1945), and the majority of the proposed methods assume access to first-order (FO) information, i.e. gradients, to find or approximate robust solutions (Nesterov, 2007; Gidel et al., 2017; Hamedani et al., 2018; Qian et al., 2019; Rafique et al., 2018; Sanjabi et al., 2018b; Lu et al., 2019; Nouiehed et al., 2019; Lu et al., 2019; Jin et al., 2019). In this paper, we focus on *design* and *analysis* of *black-box (gradient-free)* min-max optimization methods, where gradients are neither symbolically nor numerically available, or they are tedious to compute (Conn et al., 2009). Our study is particularly motivated by the design of data poisoning and evasion adversarial attacks from black-box machine learning (ML) or deep learning (DL) systems, whose internal configuration and operating mechanism are unknown to adversaries. The extension of min-max optimization from the FO domain to the gradient-free regime is challenging since the solver suffers from uncertainties in both black-box objective functions and optimization procedure and do not scale well to high-dimensional problems.

We develop a provable and unified black-box min-max stochastic optimization method by integrating a *query-efficient* randomized zeroth-order (ZO) gradient estimator with a *computation-efficient* alternating gradient descent-ascent framework, where the former requires a small number of function queries to build a gradient estimate, and the latter needs just one-step descent/ascent update. Recently, ZO optimization has attracted increasing attention in solving ML/DL problems. For example, ZO optimization serves as a powerful and practical tool for generation of black-box adversarial examples

to evaluate the adversarial robustness of ML/DL models (Chen et al., 2017; Ilyas et al., 2018; Tu et al., 2018; Ilyas et al., 2019). ZO optimization can also help to solve automated ML problems, where the gradients with respect to ML pipeline configuration parameters are intractable (Aggarwal et al., 2019). Furthermore, ZO optimization provides computationally-efficient alternatives of high-order optimization methods for solving complex ML/DL tasks, e.g., robust training by leveraging input gradient or curvature regularization (Finlay & Oberman, 2019; Moosavi-Dezfooli et al., 2019), model-agnostic meta-learning (Fallah et al., 2019), network control and management (Chen & Giannakis, 2018), and data processing in high dimension (Liu et al., 2018b). Other recent applications include generating model-agnostic contrastive explanations (Dhurandhar et al., 2019) and escaping saddle points (Flokas et al., 2019). Current studies (Ghadimi & Lan, 2013; Nesterov & Spokoiny, 2015; Duchi et al., 2015; Ghadimi et al., 2016; Shamir, 2017; Liu et al., 2019) suggested that ZO methods typically agree with the iteration complexity of FO methods but encounter a slowdown factor up to a small-degree polynomial of the problem dimensionality. To the best of our knowledge, it was an open question whether any convergence rate analysis can be established for black-box min-max optimization.

**Contribution.**     We summarize our contributions as follows. (***i***) We first identify a class of black-box attack and robust learning problems which turn out to be min-max black-box optimization problems. (***ii***) We propose a scalable and principled framework (ZO-Min-Max) for solving constrained min-max saddle point problems under both one-sided and two-sided black-box objective functions. Here the one-sided setting refers to the scenario where only the outer minimization problem is black-box. (***iii***) We provide a novel convergence analysis characterizing the number of objective function evaluations required to attain locally robust solution to black-box min-max problems with nonconvex outer minimization and strongly concave inner maximization. Our analysis handles stochasticity in both objective function and ZO gradient estimator, and shows that ZO-Min-Max yields $\mathcal{O}(1/T + 1/b + d/q)$ convergence rate, where $T$ is number of iterations, $b$ is mini-batch size, $q$ is number of random direction vectors used in ZO gradient estimation, and $d$ is number of optimization variables. (***iv***) We demonstrate the effectiveness of our proposal in practical data poisoning and evasion attack generation problems.[1]

## 2    RELATED WORK

**FO min-max optimization.**     Gradient-based methods have been applied with celebrated success to solve min-max problems such as robust learning (Qian et al., 2019), generative adversarial networks (GANs) (Sanjabi et al., 2018a), adversarial training (Al-Dujaili et al., 2018b; Madry et al., 2017a), and robust adversarial attack generation (Wang et al., 2019b). Some of FO methods are motivated by theoretical justifications based on Danskin's theorem (Danskin, 1966), which implies that the negative of the gradient of the outer minimization problem at inner maximizer is a descent direction (Madry et al., 2017a). Convergence analysis of other FO min-max methods has been studied under different problem settings, e.g., (Lu et al., 2019; Qian et al., 2019; Rafique et al., 2018; Sanjabi et al., 2018b; Nouiehed et al., 2019). It was shown in (Lu et al., 2019) that a deterministic FO min-max algorithm has $\mathcal{O}(1/T)$ convergence rate. In (Qian et al., 2019; Rafique et al., 2018), stochastic FO min-max methods have also been proposed, which yield the convergence rate in the order of $\mathcal{O}(1/\sqrt{T})$ and $\mathcal{O}(1/T^{1/4})$, respectively. However, these works were restricted to unconstrained optimization at the minimization side. In (Sanjabi et al., 2018b), noncovnex-concave min-max problems were studied, but the proposed analysis requires solving the maximization problem only up to some small error. In (Nouiehed et al., 2019), the $\mathcal{O}(1/T)$ convergence rate was proved for nonconvex-nonconcave min-max problems under Polyak- Łojasiewicz conditions. Different from the aforementioned FO settings, ZO min-max stochastic optimization suffers randomness from both stochastic sampling in objective function and ZO gradient estimation, and this randomness would be coupled in alternating gradient descent-descent steps and thus make it more challenging in convergence analysis.

**Gradient-free min-max optimization.**     In the black-box setup, coevolutionary algorithms were used extensively to solve min-max problems (Herrmann, 1999; Schmiedlechner et al., 2018). However, they may oscillate and never converge to a solution due to pathological behaviors such as *focusing* and *relativism* (Watson & Pollack, 2001). Fixes to these issues have been proposed and analyzed—e.g.,

---

[1]Source code will be released.

asymmetric fitness (Jensen, 2003; Branke & Rosenbusch, 2008). In (Al-Dujaili et al., 2018c), the authors employed an evolution strategy as an unbiased approximate for the descent direction of the outer minimization problem and showed empirical gains over coevlutionary techniques, albeit without any theoretical guarantees. Min-max black-box problems can also be addressed by resorting to direct search and model-based descent and trust region methods (Audet & Hare, 2017; Larson et al., 2019; Rios & Sahinidis, 2013). However, these methods lack convergence rate analysis and are difficult to scale to high-dimensional problems. For example, the off-the-shelf model-based solver COBYLA only supports problems with $2^{16}$ variables at maximum in SciPy Python library (Jones et al., 2001), which is even smaller than the size of a single ImageNet image. The recent work (Bogunovic et al., 2018) proposed a robust Bayesian optimization (BO) algorithm and established a theoretical lower bound on the required number of the min-max objective evaluations to find a near-optimal point. However, BO approaches are often tailored to low-dimensional problems and its computational complexity prohibits scalable application. From a game theory perspective, the min-max solution for some problems correspond to the Nash equilibrium between the outer minimizer and the inner maximizer, and hence black-box Nash equilibria solvers can be used (Picheny et al., 2019; Al-Dujaili et al., 2018a). This setup, however, does not always hold in general. Our work contrasts with the above lines of work in designing and analyzing black-box min-max techniques that are *both* scalable and theoretically grounded.

## 3 PROBLEM SETUP

In this section, we define the black-box min-max problem and briefly motivate its applications. By *min-max*, we mean that the problem is a composition of inner maximization and outer minimization of the objective function $f$. By *black-box*, we mean that the objective function $f$ is only accessible via point-wise functional evaluations. Mathematically, we have

$$\min_{\mathbf{x} \in \mathcal{X}} \max_{\mathbf{y} \in \mathcal{Y}} \quad f(\mathbf{x}, \mathbf{y}) \tag{1}$$

where $\mathbf{x}$ and $\mathbf{y}$ are optimization variables, $f$ is a differentiable objective function, and $\mathcal{X} \subset \mathbb{R}^{d_x}$ and $\mathcal{Y} \subset \mathbb{R}^{d_y}$ are compact convex sets. For ease of notation, let $d_x = d_y = d$. In (1), the objective function $f$ could represent either a deterministic loss or stochastic loss $f(\mathbf{x}, \mathbf{y}) = \mathbb{E}_{\boldsymbol{\xi} \sim p}[f(\mathbf{x}, \mathbf{y}; \boldsymbol{\xi})]$, where $\boldsymbol{\xi}$ is a random variable following the distribution $p$. In this paper, we consider the stochastic variant in (1).

We focus on two *black-box* scenarios in which gradients (or stochastic gradients under randomly sampled $\boldsymbol{\xi}$) of $f$ w.r.t. $\mathbf{x}$ or $\mathbf{y}$ are not accessed.

(a) *One-sided black-box*: $f(\mathbf{x}, \mathbf{y})$ is a white box w.r.t. $\mathbf{y}$ but a black box w.r.t. $\mathbf{x}$.

(b) *Two-sided black-box*: $f(\mathbf{x}, \mathbf{y})$ is a black box w.r.t. both $\mathbf{x}$ and $\mathbf{y}$.

**Motivation of setup (a) and (b).** Both setups are well motivated from the design of *black-box adversarial attacks*. The formulation of the *one-sided black-box* min-max problem corresponds to a particular type of attack, known as *black-box ensemble evasion attack*, where the attacker generates adversarial examples (i.e., crafted examples with slight perturbations for misclassification at the testing phase) and optimizes its worst-case performance against an *ensemble* of black-box classifiers and/or example classes. The formulation of *two-sided black-box* min-max problem represents another type of attack at the training phase, known as *black-box poisoning attack*, where the attacker deliberately influences the training data (by injecting poisoned samples) to manipulate the results of a black-box predictive model.

Although problems of designing ensemble evasion attack (Liu et al., 2016; 2018a; Wang et al., 2019b) and data poisoning attack (Jagielski et al., 2018; Wang et al., 2019a) have been studied in the literature, most of them assumed that the adversary has the *full* knowledge of the target ML model, leading to an impractical *white-box* attack setting. By contrast, we provide a solution to min-max attack generation under *black-box* ML models. We refer readers to Section 6 for further discussion and demonstration of our framework on these problems.

## 4 ZO-MIN-MAX: A FRAMEWORK FOR BLACK-BOX MIN-MAX OPTIMIZATION

Our interest is in a scalable and theoretically principled framework for black-box min-max problems of the form (1). To this end, we first introduce a randomized gradient estimator that only requires a few number of point-wise function evaluations. Based on that, we then propose a ZO alternating projected gradient method to solve (1) under both one-sided and two-sided black-box setups.

**Randomized gradient estimator.** In the ZO setting, we adopt a randomized gradient estimator to estimate the gradient of a function with the *generic* form $h(\mathbf{x}) := \mathbb{E}_{\boldsymbol{\xi}}[h(\mathbf{x};\boldsymbol{\xi})]$ (Liu et al., 2019; Gao et al., 2014),

$$\widehat{\nabla}_{\mathbf{x}} h(\mathbf{x}) = \frac{1}{bq} \sum_{j \in \mathcal{I}} \sum_{i=1}^{q} \frac{d[h(\mathbf{x} + \mu \mathbf{u}_i; \boldsymbol{\xi}_j) - h(\mathbf{x};\boldsymbol{\xi}_j)]}{\mu} \mathbf{u}_i, \tag{2}$$

where $d$ is number of variables, $\mathcal{I}$ denotes the mini-batch set of $b$ *i.i.d.* stochastic samples $\{\boldsymbol{\xi}_j\}_{j=1}^{b}$, $\{\mathbf{u}_i\}_{i=1}^{q}$ are $q$ *i.i.d.* random direction vectors drawn uniformly from the unit sphere, and $\mu > 0$ is a smoothing parameter. We note that the ZO gradient estimator (2) involves randomness from both stochastic sampling w.r.t. $\mathbf{u}_i$ as well as the random direction sampling w.r.t. $\boldsymbol{\xi}_j$. It is known from (Gao et al., 2014, Lemma 2) that $\widehat{\nabla}_{\mathbf{x}} h(\mathbf{x})$ provides an unbiased estimate of the gradient of the smoothing function of $h$ rather than the true gradient of $h$. Here the smoothing function of $h$ is defined by $h_\mu(\mathbf{x}) = \mathbb{E}_{\mathbf{v}}[h(\mathbf{x} + \mu \mathbf{v})]$, where $\mathbf{v}$ follows the uniform distribution over the unit Euclidean ball. Besides the bias, we provide an upper bound on the variance of (2) in Lemma 1.

**Lemma 1.** *Suppose that for all $\boldsymbol{\xi}$, $h(\mathbf{x};\boldsymbol{\xi})$ has $L_h$ Lipschitz continuous gradients and the gradient of $h(\mathbf{x};\boldsymbol{\xi})$ is upper bounded as $\|\nabla_{\mathbf{x}} h(\mathbf{x};\boldsymbol{\xi})\|_2^2 \leq \eta^2$ at $\mathbf{x} \in \mathbb{R}^d$. Then $\mathbb{E}\left[\widehat{\nabla}_{\mathbf{x}} h(\mathbf{x})\right] = \nabla_{\mathbf{x}} h_\mu(\mathbf{x})$,*

$$\mathbb{E}\left[\|\widehat{\nabla}_{\mathbf{x}} h(\mathbf{x}) - \nabla_{\mathbf{x}} h_\mu(\mathbf{x})\|_2^2\right] \leq \frac{2\eta^2}{b} + \frac{4d\eta^2 + \mu^2 L_h^2 d^2}{q} := \sigma^2(L_h, \mu, b, q, d), \tag{3}$$

*where the expectation is taken over all randomness.*

**Proof:** See Appendix A.2. □

In Lemma 1, if we choose $\mu \leq 1/\sqrt{d}$, then the variance bound is given by $\mathcal{O}(1/b + d/q)$. In our problem setting (1), the ZO gradients $\widehat{\nabla}_{\mathbf{x}} f(\mathbf{x}, \mathbf{y})$ and $\widehat{\nabla}_{\mathbf{y}} f(\mathbf{x}, \mathbf{y})$ follow the generic form of (2) by fixing $\mathbf{y}$ and letting $h(\cdot) := f(\cdot, \mathbf{y})$ or by fixing $\mathbf{x}$ and letting $h(\cdot) := f(\mathbf{x}, \cdot)$, respectively.

**Algorithmic framework.** To solve problem (1), we alternatingly perform ZO projected gradient descent/ascent method for updating $\mathbf{x}$ and $\mathbf{y}$. Specifically, for one-sided ZO min-max optimization, the ZO projected gradient descent (ZO-PGD) over $\mathbf{x}$ yields

$$\mathbf{x}^{(t)} = \text{proj}_{\mathcal{X}}\left(\mathbf{x}^{(t-1)} - \alpha \widehat{\nabla}_{\mathbf{x}} f\left(\mathbf{x}^{(t-1)}, \mathbf{y}^{(t-1)}\right)\right), \tag{4}$$

where $t$ is the iteration index, $\widehat{\nabla}_{\mathbf{x}} f$ denotes the ZO gradient estimate of $f$ w.r.t. $\mathbf{x}$, $\alpha > 0$ is the learning rate at the $\mathbf{x}$-minimization step, and $\text{proj}_{\mathcal{X}}(\mathbf{a})$ signifies the projection of $\mathbf{a}$ onto $\mathcal{X}$, given by the solution to the problem $\min_{\mathbf{x} \in \mathcal{X}} \|\mathbf{x} - \mathbf{a}\|_2^2$. For two-sided ZO min-max optimization, in addition to (4), our update on $\mathbf{y}$ obeys the ZO projected gradient ascent (ZO-PGA)

$$\mathbf{y}^{(t)} = \text{proj}_{\mathcal{Y}}\left(\mathbf{y}^{(t-1)} + \beta \widehat{\nabla}_{\mathbf{y}} f\left(\mathbf{x}^{(t)}, \mathbf{y}^{(t-1)}\right)\right), \tag{5}$$

where $\beta > 0$ is the learning rate at the $\mathbf{y}$-maximization step. The proposed method is named as *ZO-Min-Max*; see Algorithm 1.

**Why estimates gradient rather than distribution of function values?** Besides ZO optimization using random gradient estimates, the black-box min-max problem (1) can also be solved using the Bayesian optimization (BO) approach, e.g., (Bogunovic et al., 2018; Al-Dujaili et al., 2018a). The core idea of BO is to approximate the objective function as a Gaussian process (GP) learnt from the history of function values at queried points. Based on GP, the solution to problem (1) is then updated by maximizing a certain reward function, known as acquisition function. The advantage of BO is

its mild requirements on the setting of black-box problems, e.g., at the absence of differentiability. However, BO usually does not scale beyond low-dimensional problems since learning the accurate GP model and solving the acquisition problem takes intensive computation cost per iteration. By contrast, our proposed method is more efficient, and mimics the first-order method by just using the random gradient estimate (2) as the descent/ascent direction. In Figure A1, we compare ZO-Min-Max with the BO based STABLEOPT algorithm proposed by (Bogunovic et al., 2018) through a toy example shown in (Bogunovic et al., 2018, Sec. 5). As we can see, ZO-Min-Max not only achieves more accurate solution but also requires less computation time. We refer readers to Appendix B for details.

**Technical challenges in convergence analysis.**
The convergence analysis of ZO-Min-Max is more challenging than the case of FO min-max algorithms. Besides the inaccurate estimate of the gradient, the stochasticity of the estimator makes the convergence analysis sufficiently different from the FO deterministic case (Lu et al., 2019; Qian et al., 2019), since the errors in minimization and maximization are coupled as the algorithm proceeds.

---

**Algorithm 1** ZO-Min-Max to solve problem (1)

---

1: **Input:** given $\mathbf{x}^{(0)}$ and $\mathbf{y}^{(0)}$, learning rates $\alpha$ and $\beta$, the number of random directions $q$, and the possible mini-batch size $b$ for stochastic optimization
2: **for** $t = 1, 2, \ldots, T$ **do**
3:     $\mathbf{x}$-step: perform ZO-PGD (4)
4:     $\mathbf{y}$-step:
5:     **if** $f(\mathbf{x}^{(t)}, \mathbf{y})$ is black box w.r.t. $\mathbf{y}$ **then**
6:         perform ZO-PGA (5)
7:     **else**
8:         perform PGA using $\nabla_{\mathbf{y}} f(\mathbf{x}^{(t)}, \mathbf{y}^{(t-1)})$ as ascent direction in (5)
9:     **end if**
10: **end for**

---

Moreover, the conventioanl analysis of ZO optimization for single-objective problems cannot directly be applied to ZO-Min-Max. Even at the one-sided black-box setting, ZO-Min-Max conducts alternating optimization using one-step ZO-PGD and PGA with respect to $\mathbf{x}$ and $\mathbf{y}$ respectively. This is different from a reduced ZO optimization problem with respect to $\mathbf{x}$ only by solving problem $\min_{\mathbf{x} \in \mathcal{X}} h(\mathbf{x}) := \min_{\mathbf{y} \in \mathcal{Y}} f(\mathbf{x}, \mathbf{y})$, which requires the algorithm to obtain the solution to $\min_{\mathbf{y} \in \mathcal{Y}} f(\mathbf{x}, \mathbf{y})$ at a given $\mathbf{x}$ (when querying $h(\mathbf{x})$ for a ZO gradient estimation). However, this process is usually non-trivial or computationally intensive.

In particular, one key difficulty stems from the alternating algorithmic structure (namely, primal-dual framework) as the problem is in the min-max form, which leads to opposite optimization directions (minimization vs maximization) over variable $\mathbf{x}$ and $\mathbf{y}$ respectively. Even applying ZO optimization only to one side, it needs to quantify the effect of ZO gradient estimation on the descent over both $\mathbf{x}$ and $\mathbf{y}$. We provide a detailed convergence analysis of ZO-Min-Max in the next section.

## 5 CONVERGENCE ANALYSIS

We begin by elaborating on assumptions and notations used in analyzing the convergence of ZO-Min-Max (Algorithm 1).

**A1**: In (1), $f(\mathbf{x}, \mathbf{y})$ is continuously differentiable, and is strongly concave w.r.t. $\mathbf{y}$ with parameter $\gamma > 0$, namely, given $\mathbf{x} \in \mathcal{X}$, $f(\mathbf{x}, \mathbf{y}_1) \leq f(\mathbf{x}, \mathbf{y}_2) + \nabla_{\mathbf{y}} f(\mathbf{x}, \mathbf{y}_2)^T (\mathbf{y}_1 - \mathbf{y}_2) - \frac{\gamma}{2} \|\mathbf{y}_1 - \mathbf{y}_2\|^2$ for all points $\mathbf{y}_1, \mathbf{y}_2 \in \mathcal{Y}$. And $f$ is lower bounded by a finite number $f^*$ and has bounded gradients $\|\nabla_{\mathbf{x}} f(\mathbf{x}, \mathbf{y}; \boldsymbol{\xi})\| \leq \eta^2$ and $\|\nabla_{\mathbf{y}} f(\mathbf{x}, \mathbf{y}; \boldsymbol{\xi})\| \leq \eta^2$ for stochastic optimization with $\boldsymbol{\xi} \sim p$. Here $\|\cdot\|$ denotes the $\ell_2$ norm. The constraint sets $\mathcal{X}, \mathcal{Y}$ are convex and bounded with diameter $R$.

**A2**: $f(\mathbf{x}, \mathbf{y})$ has Lipschitz continuous gradients, i.e., there exists $L_x, L_y > 0$ such that $\|\nabla_{\mathbf{x}} f(\mathbf{x}_1, \mathbf{y}) - \nabla_{\mathbf{x}} f(\mathbf{x}_2, \mathbf{y})\| \leq L_x \|\mathbf{x}_1 - \mathbf{x}_2\|$ for $\forall \mathbf{x}_1, \mathbf{x}_2 \in \mathcal{X}$, and $\|\nabla_{\mathbf{y}} f(\mathbf{x}_1, \mathbf{y}) - \nabla_{\mathbf{y}} f(\mathbf{x}_2, \mathbf{y})\| \leq L_y \|\mathbf{x}_1 - \mathbf{x}_2\|$ and $\|\nabla_{\mathbf{y}} f(\mathbf{x}, \mathbf{y}_1) - \nabla_{\mathbf{y}} f(\mathbf{x}, \mathbf{y}_2)\| \leq L_y \|\mathbf{y}_1 - \mathbf{y}_2\|$ for $\forall \mathbf{y}_1, \mathbf{y}_2 \in \mathcal{Y}$.

We note that **A1** and **A2** are required for analyzing the convergence of ZO-Min-Max. They were used even for the analysis of first-order min-max optimization methods (Lu et al., 2019; Nouiehed et al., 2019) and first-order methods for nonconvex optimization with a single objective function (Chen et al., 2019; Ward et al., 2019). In **A1**, the strongly concavity of $f(\mathbf{x}, \mathbf{y})$ with respect to $\mathbf{y}$ holds for applications such as robust learning over multiple domains (Qian et al., 2019), and adversarial attack generation that will be introduced in Section 6. In **A2**, the assumption of smoothness

(namely, Lipschitz continuous gradient) is required to quantify the descent of the alternating projected stochastic gradient descent-ascent method. Even for single-objective non-convex optimization, e.g., (Chen et al., 2019; Bernstein et al., 2018), **A2** is needed in analysis. For clarity, we also summarize the problem and algorithmic parameters used in our convergence analysis in Table A1 of Appendix.

We measure the convergence of ZO-Min-Max by the proximal gradient (Lu et al., 2019; Ghadimi et al., 2016),

$$\mathcal{G}(\mathbf{x}, \mathbf{y}) = \left[ \begin{array}{c} (1/\alpha)\left(\mathbf{x} - \text{proj}_{\mathcal{X}}(\mathbf{x} - \alpha\nabla_{\mathbf{x}}f(\mathbf{x}, \mathbf{y}))\right) \\ (1/\beta)\left(\mathbf{y} - \text{proj}_{\mathcal{Y}}(\mathbf{y} + \beta\nabla_{\mathbf{y}}f(\mathbf{x}, \mathbf{y}))\right) \end{array} \right], \tag{6}$$

where $(\mathbf{x}, \mathbf{y})$ is a first-order stationary point of (1) iff $\|\mathcal{G}(\mathbf{x}, \mathbf{y})\| = 0$.

In what follows, we delve into our convergence analysis. First, Lemma 2 shows the descent property of ZO-PGD at the $\mathbf{x}$-minimization step in Algorithm 1.

**Lemma 2.** *(Descent lemma in minimization) Under **A1**-**A2**, let $(\mathbf{x}^{(t)}, \mathbf{y}^{(t)})$ be a sequence generated by Algorithm 1. When $f(\mathbf{x}, \mathbf{y})$ is black-box w.r.t. $\mathbf{x}$, then we have following descent property w.r.t. $\mathbf{x}$:*

$$\mathbb{E}[f(\mathbf{x}^{(t+1)}, \mathbf{y}^{(t)})] \leq \mathbb{E}[f(\mathbf{x}^{(t)}, \mathbf{y}^{(t)})] - \left(\frac{1}{\alpha} - \frac{L_x}{2}\right)\mathbb{E}\|\Delta_{\mathbf{x}}^{(t+1)}\|^2 + \alpha\sigma_x^2 + L_x\mu^2 \tag{7}$$

*where $\Delta_{\mathbf{x}}^{(t)} := \mathbf{x}^{(t)} - \mathbf{x}^{(t-1)}$, and $\sigma_x^2 := \sigma^2(L_x, \mu, b, q, d)$ defined in (3).*

**Proof**: See Appendix A.3.1. $\qquad\qquad\qquad\qquad\qquad\qquad\qquad\qquad\qquad\qquad\qquad\qquad\quad\square$

It is clear from Lemma 2 that updating $\mathbf{x}$ leads to the reduced objective value when choosing a small learning rate $\alpha$. However, ZO gradient estimation brings in additional errors in terms of $\alpha\sigma_x^2$ and $L_x\mu^2$, where the former is induced by the variance of gradient estimates in (3) and the latter is originated from bounding the distance between $f$ and its smoothing version; see (25) in Appendix A.3.

**Convergence rate of ZO-Min-Max by performing PGA.** We next investigate the convergence of ZO-Min-Max when FO PGA is used at the $\mathbf{y}$-maximization step (Line 8 of Algorithm 1) for solving one-sided black-box optimization problems.

**Lemma 3.** *(Descent lemma in maximization) Under **A1**-**A2**, let $(\mathbf{x}^{(t)}, \mathbf{y}^{(t)})$ be a sequence generated by Algorithm 1 and define the potential function as*

$$\mathcal{P}(\mathbf{x}^{(t)}, \mathbf{y}^{(t)}, \Delta_{\mathbf{y}}^{(t)}) = \mathbb{E}[f(\mathbf{x}^{(t)}, \mathbf{y}^{(t)})] + \frac{4 + 4\beta^2 L_y^2 - 7\beta\gamma}{2\beta^2\gamma}\mathbb{E}\|\Delta_{\mathbf{y}}^{(t)}\|^2, \tag{8}$$

*where $\Delta_{\mathbf{y}}^{(t)} := \mathbf{y}^{(t)} - \mathbf{y}^{(t-1)}$. When $f(\mathbf{x}, \mathbf{y})$ is black-box w.r.t. $\mathbf{x}$ and white-box w.r.t. $\mathbf{y}$, then we have the following descent property w.r.t. $\mathbf{y}$:*

$$\mathcal{P}(\mathbf{x}^{(t+1)}, \mathbf{y}^{(t+1)}, \Delta_{\mathbf{y}}^{(t+1)}) \leq \mathcal{P}(\mathbf{x}^{(t+1)}, \mathbf{y}^{(t)}, \Delta_{\mathbf{y}}^{(t)})$$
$$- \left(\frac{1}{2\beta} - \frac{2L_y^2}{\gamma}\right)\mathbb{E}\|\Delta_{\mathbf{y}}^{(t+1)}\|^2 + \left(\frac{2}{\gamma^2\beta} + \frac{\beta}{2}\right)L_x^2\mathbb{E}\|\Delta_{\mathbf{x}}^{(t+1)}\|^2, \tag{9}$$

**Proof**: See Appendix A.3.2. $\qquad\qquad\qquad\qquad\qquad\qquad\qquad\qquad\qquad\qquad\qquad\qquad\quad\square$

It is shown from (9) that when $\beta$ is small enough, then the term $(1/(2\beta) - 2L_y^2/\gamma)\mathbb{E}\|\Delta_{\mathbf{y}}^{(t+1)}\|^2$ will give some descent of the potential function after performing PGA, while the last term in (9) will give some ascent to the potential function. However, such a quantity will be compensated by the descent of the objective function in the minimization step shown by Lemma 2. Combining Lemma 2 and Lemma 3, we obtain the convergence rate of ZO-Min-Max in Theorem 1.

**Theorem 1.** *Suppose that **A1**-**A2** hold, the sequence $(\mathbf{x}^{(t)}, \mathbf{y}^{(t)})$ over $T$ iterations is generated by Algorithm 1 in which learning rates satisfy $\beta < 1/(4L_y^2)$ and $\alpha \leq \min\{1/L_x, 1/(L_x/2 + 2L_x^2/(\gamma^2\beta) + \beta L_x^2/2)\}$. When $f(\mathbf{x}, \mathbf{y})$ is black-box w.r.t. $\mathbf{x}$ and white-box w.r.t. $\mathbf{y}$, the convergence rate of ZO-Min-Max under a uniformly and randomly picked $(\mathbf{x}^{(r)}, \mathbf{y}^{(r)})$ from $\{(\mathbf{x}^{(t)}, \mathbf{y}^{(t)})\}_{t=1}^T$ is given by*

$$\mathbb{E}\|\mathcal{G}(\mathbf{x}^{(r)}, \mathbf{y}^{(r)})\|^2 \leq \frac{c}{\zeta}\frac{(\mathcal{P}_1 - f^* - \nu R^2)}{T} + \frac{c\alpha\sigma_x^2}{\zeta} + \frac{cL_x\mu^2}{\zeta} \tag{10}$$

*where $\zeta$ is a constant independent on the parameters $\mu$, $b$, $q$, $d$ and $T$, $\mathcal{P}_t := \mathcal{P}(\mathbf{x}^{(t)}, \mathbf{y}^{(t)}, \Delta_{\mathbf{y}}^{(t)})$ given by (8), $c = \max\{L_x + 3/\alpha, 3/\beta\}$, $\nu = \min\{4 + 4\beta^2 L_y^2 - 7\beta\gamma, 0\}/(2\beta^2\gamma)$, $\sigma_x^2$ is variance bound of ZO gradient estimate given in (7), and $f^*$, $R$, $\gamma$, $L_x$ and $L_y$ have been defined in A1-A2.*

**Proof**: See Appendix A.3.3. □

To better interpret Theorem 1, we begin by clarifying the parameters involved in our convergence rate (10). First, the parameter $\zeta$ appears in the denominator of the derived convergence error. However, $\zeta$ has a non-trivial lower bound given appropriate learning rates $\alpha$ and $\beta$ (see **Remark 1** that we will show later). Second, the parameter $c$ is inversely proportional to $\alpha$ and $\beta$. Thus, to guarantee the constant effect of the ratio $c/\xi$, it is better not to set these learning rates too small; see a specification in **Remark 1-2**. Third, the parameter $\nu$ is non-negative and appears in terms of $-\nu R^2$, thus, it will not make convergence rate worse. Fourth, $\mathcal{P}_1$ is the initial value of the potential function (8). By setting an appropriate learning rate $\beta$ (e.g., following **Remark 2**), $\mathcal{P}_1$ is then upper bounded by a constant determined by the initial value of the objective function, the distance of the first two updates, Lipschitz constant $L_y$ and strongly concave parameter $\gamma$. We next provide **Remarks 1-3** on Theorem 1.

**Remark 1.** Recall that $\zeta = \min\{c_1, c_2\}$ (Appendix B.2.3), where $c_1 = 1/(2\beta) - 2L_y^2/\gamma$ and $c_2 = \frac{1}{\alpha} - (\frac{L_x}{2} + \frac{2L_x^2}{\gamma^2\beta} + \frac{\beta L_x^2}{2})$. Given the fact that $L_x$ and $L_y$ are Lipschitz constants and $\gamma$ is the strongly concavity constant, a proper lower bound of $\zeta$ thus relies on the choice of the learning rates $\alpha$ and $\beta$. By setting $\beta \le \frac{\gamma}{8L_y^2}$ and $\alpha \le 1/(L_x + \frac{4L_x^2}{\gamma^2\beta} + \beta L_x^2)$, it is easy to verify that $c_1 \ge \frac{2L_y^2}{\gamma}$ and $c_2 \ge \frac{L_x}{2} + \frac{2L_x^2}{\gamma^2\beta} + \frac{\beta L_x^2}{2} \ge \frac{L_x}{2} + \frac{2L_x^2}{\gamma}$. Thus, we obtain that $\zeta \ge \min\{\frac{2L_y^2}{\gamma}, \frac{2L_x^2}{\gamma} + \frac{L_x}{2}\}$. This justifies that $\zeta$ has a non-trivial lower bound, which will not make the convergence error bound (10) vacuous (although the bound has not been optimized over $\alpha$ and $\beta$).

**Remark 2.** It is not wise to set learning rates $\alpha$ and $\beta$ to extremely small values since $c$ is *inversely proportional* to $\alpha$ and $\beta$. Thus, we typically choose $\beta = \frac{\gamma}{8L_y^2}$ and $\alpha = 1/(L_x + \frac{4L_x^2}{\gamma^2\beta} + \beta L_x^2)$ in Remark 1 to guarantee the constant effect of $c/\zeta$.

**Remark 3.** By setting $\mu \le \min\{1/\sqrt{d}, 1/\sqrt{T}\}$, we obtain $\sigma_x^2 = \mathcal{O}(1/b + d/q)$ from Lemma 1, and Theorem 1 implies that ZO-Min-Max yields $\mathcal{O}(1/T + 1/b + d/q)$ convergence rate for one-sided black-box optimization. Compared to the FO rate $\mathcal{O}(1/T)$ (Lu et al., 2019; Sanjabi et al., 2018a), ZO-Min-Max converges only to a neighborhood of stationary points with $\mathcal{O}(1/T)$ rate, where the size of the neighborhood is determined by the mini-batch size $b$ and the number of random direction vectors $q$ used in ZO gradient estimation. It is also worth mentioning that such a stationary gap may exist even in the FO/ZO projected stochastic gradient descent for solving single-objective minimization problems (Ghadimi et al., 2016).

As shown in **Remark 3**, ZO-Min-Max could result in a stationary gap. A large mini-batch size $b$ or number of random direction vectors $q$ can improve its iteration complexity. However, this requires $O(bq)$ times more function queries per iteration from (2). It implies the tradeoff between iteration complexity and function query complexity in ZO optimization.

**Convergence rate of ZO-Min-Max by performing ZO-PGA.** We now focus on the convergence analysis of ZO-Min-Max when ZO PGA is used at the $\mathbf{y}$-maximization step (Line 6 of Algorithm 1) for two-sided black-box optimization problems.

**Lemma 4.** *(Descent lemma in maximization) Under A1-A2, let $(\mathbf{x}^{(t)}, \mathbf{y}^{(t)})$ be a sequence generated by Algorithm 1 and define the potential function as*

$$\mathcal{P}'(\mathbf{x}^{(t)}, \mathbf{y}^{(t)}, \Delta_{\mathbf{y}}^{(t)}) = \mathbb{E}[f(\mathbf{x}^{(t)}, \mathbf{y}^{(t)})] + \frac{4 + 4(3L_y^2 + 2)\beta^2 - 7\beta\gamma}{\beta^2\gamma} \mathbb{E}\|\Delta_{\mathbf{y}}^{(t)}\|^2. \quad (11)$$

*When function $f(\mathbf{x}, \mathbf{y})$ is black-box w.r.t. both $\mathbf{x}$ and $\mathbf{y}$, we have the following descent w.r.t. $\mathbf{y}$:*

$$\mathcal{P}'(\mathbf{x}^{(t+1)}, \mathbf{y}^{(t+1)}, \Delta_{\mathbf{y}}^{(t+1)}) \le \mathcal{P}'(\mathbf{x}^{(t+1)}, \mathbf{y}^{(t)}, \Delta_{\mathbf{y}}^{(t)}) - \left(\frac{1}{2\beta} - \frac{6L_y^2 + 4}{\gamma}\right) \mathbb{E}\|\Delta_{\mathbf{y}}^{(t+1)}\|^2$$

$$+ \left(\frac{6L_x^2}{\gamma^2\beta} + \frac{3\beta L_x^2}{2}\right) \mathbb{E}\|\Delta_{\mathbf{x}}^{(t+1)}\|^2 + \frac{7\beta^2\gamma^2 + 28\beta\gamma + 12}{\beta\gamma^2}\sigma_y^2 + \frac{\beta\gamma + 4}{4\beta^2\gamma}\mu^2 d^2 L_y^2, \quad (12)$$

where $\sigma_y^2 := \sigma^2(L_y, \mu, b, q, d)$ given in (3).

**Proof**: See Appendix A.4.1. □

Lemma 4 is analogous to Lemma 3 by taking into account the effect of ZO gradient estimate $\widehat{\nabla}_{\mathbf{y}} f(\mathbf{x}, \mathbf{y})$ on the potential function (11). Such an effect is characterized by the terms related to $\sigma_y^2$ and $\mu^2 d^2 L_y^2$ in (12).

**Theorem 2.** *Suppose that A1-A2 hold, the sequence $(\mathbf{x}^{(t)}, \mathbf{y}^{(t)})$ over $T$ iterations is generated by Algorithm 1 in which learning rates satisfy $\beta < \gamma/(4(3L_y^2 + 2))$ and $\alpha \leq \min\{L_x, 1/(L_x/2 + 6L_x^2/(\gamma^2\beta) + 3\beta L_x^2/2)\}$. When $f(\mathbf{x}, \mathbf{y})$ is black-box w.r.t. both $\mathbf{x}$ and $\mathbf{y}$, the convergence rate of ZO-Min-Max under a uniformly and randomly picked $(\mathbf{x}^{(r)}, \mathbf{y}^{(r)})$ from $\{(\mathbf{x}^{(t)}, \mathbf{y}^{(t)})\}_{t=1}^T$ is given by*

$$\mathbb{E}\|\mathcal{G}(\mathbf{x}^{(r)}, \mathbf{y}^{(r)})\|^2 \leq \frac{c}{\zeta'}\frac{\mathcal{P}_1' - f^* - \nu' R^2}{T} + \frac{c\alpha}{\zeta'}\sigma_x^2 + \left(\frac{cb_1}{\zeta'} + d^2 L_y^2\right)\mu^2 + \left(\frac{cb_2}{\zeta'} + 2\right)\sigma_y^2,$$

*where $\zeta'$ is a constant independent on the parameters $\mu$, $b$, $q$, $d$ and $T$, $\mathcal{P}_t' := \mathcal{P}'(\mathbf{x}^{(t)}, \mathbf{y}^{(t)}, \Delta_{\mathbf{y}}^{(t)})$ in (11), $c$ has been defined in (10), $\nu' = \frac{\min\{4 + 4(3L_y^2+2)\beta^2 - 7\beta\gamma, 0\}}{\beta^2\gamma}$, $b_1 = L_x + \frac{d^2 L_y^2(4+\beta\gamma)}{4\beta^2\gamma}$ and $b_2 = \frac{7\beta^2\gamma^2 + 28\beta\gamma + 12}{\beta\gamma^2}$, $\sigma_x^2$ and $\sigma_y^2$ have been introduced in (7) and (12), and $f^*$, $R$, $\gamma$, $L_x$ and $L_y$ have been defined in A1-A2.*

**Proof**: See Appendix A.4.2. □

Following the similar argument in Remark 1 of Theorem 1, one can choose proper learning rates $\alpha$ and $\beta$ to obtain valid lower bound on $\zeta'$. However, different from Theorem 1, the convergence error shown by Theorem 2 involves an additional error term related to $\sigma_y^2$ and has worse dimension-dependence on the term related to $\mu^2$. The latter yields a more restricted choice of the smoothing parameter $\mu$: we obtain $\mathcal{O}(1/T + 1/b + d/q)$ convergence rate when $\mu \leq 1/(d\sqrt{T})$.

## 6 EXPERIMENTS

In this section, we evaluate the empirical performance of ZO-Min-Max on applications of adversarial exploration: 1) design of black-box ensemble attack against two neural networks Inception-V3 (Szegedy et al., 2016) and ResNet-50 (He et al., 2016) under ImageNet (Deng et al., 2009), and 2) design of black-box poisoning attack against a logistic regression model.

**Black-box ensemble evasion attack via universal perturbation**   We consider the scenario in which the attacker generates adversarial examples against an *ensemble* of multiple classifiers and/or image classes (Liu et al., 2016; 2018a). More formally, let $(\mathbf{z}, l)$ denote a legitimate image $\mathbf{z}$ with the true class label $l$, and $\mathbf{z}' := \mathbf{z} + \mathbf{x}$ denote an adversarial example, where $\mathbf{x}$ signifies the adversarial perturbation. Here the natural image $\mathbf{z}$ and the perturbed image $\mathbf{z} + \mathbf{x}$ are normalized to $[-0.5, 0.5]^d$. Considering $I$ classes of images (each group of images corresponding to the same class $l_i$ is denoted by $\Omega_i$) and $J$ network models, the adversary is to find the *universal perturbation* $\mathbf{x}$ across $I$ image classes and $J$ models. The proposed attack problem is given by

$$\underset{\mathbf{x}\in\mathcal{X}}{\text{minimize}} \, \underset{\mathbf{w}\in\mathcal{W}}{\text{maximize}} \quad f_1(\mathbf{x}, \mathbf{w}) := \sum_{j=1}^J \sum_{i=1}^I [w_{ij} F_{ij}(\mathbf{x}; \Omega_i, l_i)] - \lambda\|\mathbf{w} - \mathbf{1}/(IJ)\|_2^2, \tag{13}$$

where $\mathbf{x}$ and $\mathbf{w} \in \mathbb{R}^{IJ}$ are optimization variables, and $w_{ij}$ denotes the $(i, j)$th entry of $\mathbf{w}$ corresponding to the importance weight of attacking image class $i$ under neural network model $j$. In problem (13), $\mathcal{X}$ denotes the perturbation constraint, e.g., $\mathcal{X} = \{\mathbf{x} \mid \|\mathbf{x}\|_\infty \leq \epsilon, \mathbf{z} + \mathbf{x} \in [-0.5, 0.5]^d, \forall \mathbf{z} \in \cup_i \Omega_i\}$, $\mathcal{W} = \{\mathbf{w} \mid \mathbf{1}^T\mathbf{w} = 1, \mathbf{w} \geq 0\}$, $F_{ij}(\mathbf{x}; \Omega_i, l_i)$ is the attack loss for attacking the set of images at class $l_i$ under model $j$, and $\lambda > 0$ is a regularization parameter. We note that $\{F_{ij}\}$ in (13) are black-box functions w.r.t. $\mathbf{x}$ since the network models are blind to the adversary, which cannot perform back-propagation to obtain gradients. By contrast, it is a white-box and strongly concave function w.r.t. $\mathbf{w}$ once the function values of $\{F_{ij}\}$ are given. Thus, problem (13) belongs to the *one-sided black-box* optimization problem.

In our experiments, we consider $J = 2$ for Inception-V3 and ResNet-50, and $I = 2$ for two classes, each of which contains 20 images randomly selected from ImageNet (Deng et al., 2009). We also

specify the attack loss $F_{ij}$ in (13) as the C&W untargeted attack loss (Carlini & Wagner, 2017),

$$F_{ij}\left(\mathbf{x}; \Omega_i, l_i\right) = (1/|\Omega_i|) \sum_{\mathbf{z} \in \Omega_i} \max\{g_j(\mathbf{z} + \mathbf{x})_{l_i} - \max_{k \neq l_i} g_j(\mathbf{z} + \mathbf{x})_k, 0\}, \quad (14)$$

where $|\Omega_i|$ is the cardinality of the set $\Omega_i$, $g_j(\mathbf{z} + \mathbf{x})_k$ denotes the prediction score of class $k$ given the input $\mathbf{z} + \mathbf{x}$ using model $j$. In (13), we also set $\lambda = 5$. In Algorithm 1, we set $\alpha = 0.05, \beta = 0.01, q = 10$ and $\mu = 5 \times 10^{-3}$, and use the full batch of image samples in attack generation.

In experiment, we compare ZO-Min-Max with FO-Min-Max and ZO-Finite-Sum, where the former is the FO counterpart of Algorithm 1, and the latter is ZO-PSGD (Ghadimi et al., 2016) to minimize the finite-sum (average) loss rather than the worst-case (min-max) loss. The comparison with ZO-Finite-Sum was motivated by the previous work on designing the adversarial perturbation against model ensembles (Liu et al., 2018a) in which the averaging attack loss over multiple models was considered. Note that although ZO-Finite-Sum consider a different loss function, it is a baseline from the perspective of attack generation.

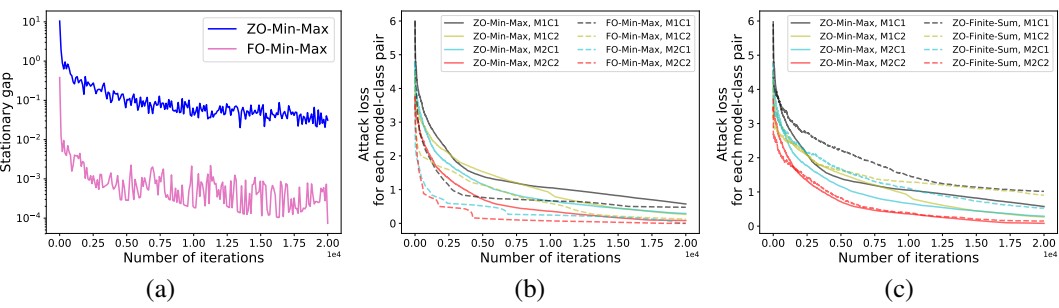

(a)        (b)        (c)

**Figure 1:** Convergence performance of ZO-Min-Max in design of black-box ensemble attack. a) Stationary gap of ZO-Min-Max vs. FO-Min-Max, b) attack loss of using ZO-Min-Max vs. FO-Min-Max, and c) attack loss of using ZO-Min-Max vs. ZO-Finite-Sum.

In Figure 1, we demonstrate the empirical convergence of ZO-Min-Max to solve problem (13) from the stationary gap $\|\mathcal{G}(\mathbf{x}, \mathbf{y})\|_2$ given in (6) and the attack loss $F_{ij}$ under each model-class pair. In Figure 1-(a), the stationary gap decreases as the iteration increases, which is consistent with the reduction in the attack loss at each M$j$C$i$. Here M and C represents network model and image class, respectively. By comparing ZO-Min-Max with FO-Min-Max in Figure 1-(b), we see that the latter yields faster convergence than the former. However, FO-Min-Max has to access the full knowledge on the target neural network for computing the gradient of individual attack losses, yielding white-box attack rather than black-box attack. In Figure 1-(c), We also compare ZO-Min-Max with ZO-Finite-Sum, where the latter minimizes the average loss $\sum_{j=1}^{J} \sum_{i=1}^{I} F_{ij}$ over all model-class combinations. As we can see, our approach significantly improves the worst-case attack performance (corresponding to M1C1). Here the worst case represents the most robust model-class pair against the attack. This suggests that ZO-Min-Max takes into account different robustness levels of model-class pairs through the design of importance weights $\mathbf{w}$. This can also be evidenced from Figure A2 in Appendix: M1C1 has the largest weight while M2C2 corresponds to the smallest weight. In Figure A3 of Appendix, we further contrast the success or failure of attacking each image using the obtained universal perturbation $\mathbf{x}$ with the attacking difficulty (in terms of required iterations for successful adversarial example) of using per-image non-universal PGD attack (Madry et al., 2017b).

**Black-box poisoning attack against logistic regression model** Let $\mathcal{D} = \{\mathbf{z}_i, t_i\}_{i=1}^n$ denote the training dataset, among which $n' \ll n$ samples are corrupted by a perturbation vector $\mathbf{x}$, leading to poisoned training data $\mathbf{z}_i + \mathbf{x}$ towards breaking the training process and thus the prediction accuracy. The poisoning attack problem is then formulated as

$$\underset{\|\mathbf{x}\|_\infty \leq \epsilon}{\text{maximize}} \underset{\boldsymbol{\theta}}{\text{minimize}} \quad f_2(\mathbf{x}, \boldsymbol{\theta}) := F_{\text{tr}}(\mathbf{x}, \boldsymbol{\theta}; \mathcal{D}_0) + \lambda\|\boldsymbol{\theta}\|_2^2, \quad (15)$$

where $\mathbf{x}$ and $\boldsymbol{\theta}$ are optimization variables, $F_{\text{tr}}(\mathbf{x}, \boldsymbol{\theta}; \mathcal{D}_0)$ denotes the training loss over model parameters $\boldsymbol{\theta}$ at the presence of data poison $\mathbf{x}$, and $\lambda > 0$ is a regularization parameter. Note that problem (15) can be written in the form of (1) with the objective function $-f_2(\mathbf{x}, \boldsymbol{\theta})$. Clearly, if $F_{\text{tr}}$ is a convex

loss (e.g., logistic regression or linear regression (Jagielski et al., 2018)), then $-f_2$ is strongly concave in $\boldsymbol{\theta}$. Since the adversary has no knowledge on the training procedure and data, $f_2(\mathbf{x}, \boldsymbol{\theta})$ is a *two-sided black-box* function. We provide more details on problem (15) in Appendix C. In Algorithm 1, unless specified otherwise we choose $b = 100$, $q = 5$, $\alpha = 0.02$, $\beta = 0.05$, and $T = 50000$. We report the empirical results *averaged* over 10 independent trials with random initialization. We compare our method with FO-Min-Max and the BO solver for robust optimization STABLEOPT (Bogunovic et al., 2018) in the data poisoning example of a relatively small problem size.

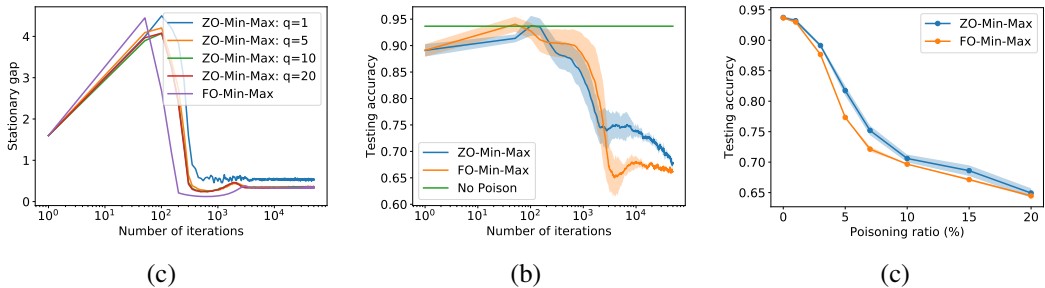

(c)   (b)   (c)

**Figure 2:** Empirical performance of ZO-Min-Max in design of poisoning attack: a) stationary gap versus iterations b) testing accuracy versus iterations (the shaded region represents variance of 10 random trials), and c) testing accuracy versus data poisoning ratio.

In Figure 2, we present the convergence performance of ZO-Min-Max to generate the data poisoning attack and validate its attack performance in terms of testing accuracy of the logistic regression model trained on the poisoned dataset. Unless specified otherwise, we set $15\%$ poisoning ratio and $\lambda = 10^{-3}$ for problem (15). We examine the sensitivity of the regularization parameter $\lambda$ in Figure A4. Figure 2-(a) shows the stationary gap defined in (6) obtained by ZO-Min-Max under different number of random direction vectors while estimating gradients (2). As we can see, a moderate choice of $q$ (e.g., $q \geq 5$ in our example) is sufficient to achieve near-optimal solution compared with FO-Min-Max. However, it suffers from a convergence bias due to the presence of stochastic sampling, consistent with Theorem 1 and 2.

Figure 2-(b) demonstrates the testing accuracy (against iterations) of the model learnt from poisoned training data, where the poisoning attack is generated by ZO-Min-Max (black-box attack) and FO-Min-Max (white-box attack). As we can see, ZO-Min-Max yields promising attacking performance comparable to FO-Min-Max. We can also see that by contrast with the testing accuracy of the clean model ($94\%$ without poison), the poisoning attack eventually reduces the testing accuracy (below $70\%$). Furthermore, in Figure 2-(c), we present the testing accuracy of the learnt model under different data poisoning ratios. As we can see, only $5\%$ poisoned training data can significantly break the testing accuracy of a well-trained model. In Figure 3, we compare ZO-Min-Max with STABLEOPT (Bogunovic et al., 2018) in terms of testing accuracy versus computation time. Fol-

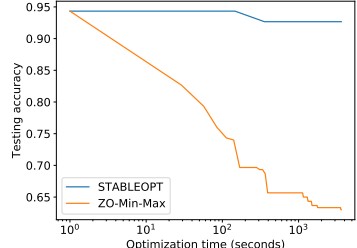

**Figure 3:** Comparison between ZO-Min-Max and STABLEOPT on testing accuracy versus optimization time.

lowing (Bogunovic et al., 2018), we present the best accuracy achieved up to the current time step. We observe that STABLEOPT is has a poorer scalability while our method reaches a data poisoning attack that induces much worse testing accuracy within $500$ seconds.

## 7 CONCLUSION

This paper addresses black-box robust optimization problems given a finite number of function evaluations. In particular, we present ZO-Min-Max: a framework of alternating, randomized gradient estimation based ZO optimization algorithm to find a first-order stationary solution to the black-box min-max problem. Under mild assumptions, ZO-Min-Max enjoys a sub-linear convergence rate. It scales to dimensions that are infeasible for recent robust solvers based on Bayesian optimization. Furthermore, we experimentally demonstrate the potential application of the framework on real-world scenarios, viz. black-box evasion and data poisoning attacks.

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

APPENDIX

## A   DETAILED CONVERGENCE ANALYSIS

### A.1   TABLE OF PARAMETERS

In Table A1, we summarize the problem and algorithmic parameters used in our convergence analysis.

**Table A1:** Summary of problem and algorithmic parameters and their descriptions.

| parameter | description |
|---|---|
| $d$ | # of optimization variables |
| $b$ | mini-batch size |
| $q$ | # of random direction vectors used in ZO gradient estimation |
| $\alpha$ | learning rate for ZO-PGD |
| $\beta$ | learning rate for ZO-PGA |
| $\gamma$ | strongly concavity parameter of $f(\mathbf{x}, \mathbf{y})$ with respect to $\mathbf{y}$ |
| $\eta$ | upper bound on the gradient norm, implying Lipschitz continuity |
| $L_x, L_y$ | Lipschitz continuous gradient constant of $f(\mathbf{x}, \mathbf{y})$ with respect to $\mathbf{x}$ and $\mathbf{y}$ respectively |
| $R$ | diameter of the compact convex set $\mathcal{X}$ or $\mathcal{Y}$ |
| $f^*$ | lower bound on the function value, implying feasibility |
| $\sigma_x^2, \sigma_y^2$ | variance of ZO gradient estimator for variable $\mathbf{x}$ and $\mathbf{y}$ respectively |

### A.2   PROOF OF LEMMA 1

Before going into the proof, let's review some preliminaries and give some definitions. Define $h_\mu(\mathbf{x}, \boldsymbol{\xi})$ to be the smoothed version of $h(\mathbf{x}, \boldsymbol{\xi})$ and since $\boldsymbol{\xi}$ models a subsampling process over a finite number of candidate functions, we can further have $h_\mu(\mathbf{x}) \triangleq \mathbb{E}_{\boldsymbol{\xi}}[h_\mu(\mathbf{x}, \boldsymbol{\xi})]$ and $\nabla_{\mathbf{x}} h_\mu(\mathbf{x}) = \mathbb{E}_{\boldsymbol{\xi}}[\nabla_{\mathbf{x}} h_\mu(\mathbf{x}, \boldsymbol{\xi})]$

Recall that in the finite sum setting when $\boldsymbol{\xi}_j$ parameterizes the $j$th function, the gradient estimator is given by

$$\widehat{\nabla}_{\mathbf{x}} h(\mathbf{x}) = \frac{1}{bq} \sum_{j \in \mathcal{I}} \sum_{i=1}^{q} \frac{d[h(\mathbf{x} + \mu \mathbf{u}_i; \boldsymbol{\xi}_j) - h(\mathbf{x}; \boldsymbol{\xi}_j)]}{\mu} \mathbf{u}_i. \tag{16}$$

where $\mathcal{I}$ is a set with $b$ elements, containing the indices of functions selected for gradient evaluation.

From standard result of the zeroth order gradient estimator, we know

$$\mathbb{E}_{\mathcal{I}} \left[ \mathbb{E}_{\mathbf{u}_i, i \in [q]} \left[ \widehat{\nabla}_{\mathbf{x}} h(\mathbf{x}) \right] \big| \mathcal{I} \right] = \mathbb{E}_{\mathcal{I}} \left[ \frac{1}{b} \sum_{j \in \mathcal{I}} \nabla_{\mathbf{x}} f_\mu(\mathbf{x}, \boldsymbol{\xi}_j) \right] = \nabla_{\mathbf{x}} h_\mu(\mathbf{x}). \tag{17}$$

Now let's go into the proof. First, we have

$$\mathbb{E} \left[ \| \widehat{\nabla}_{\mathbf{x}} h(\mathbf{x}) - \nabla_{\mathbf{x}} h_\mu(\mathbf{x}) \|_2^2 \right]$$

$$= \mathbb{E}_{\mathcal{I}} \left[ \mathbb{E}_{\mathbf{u}_i, i \in [q]} \left[ \left\| \widehat{\nabla}_{\mathbf{x}} h(\mathbf{x}) - \frac{1}{b} \sum_{j \in \mathcal{I}} \nabla_{\mathbf{x}} f_\mu(\mathbf{x}, \boldsymbol{\xi}_j) + \frac{1}{b} \sum_{j \in \mathcal{I}} \nabla_{\mathbf{x}} f_\mu(\mathbf{x}, \boldsymbol{\xi}_j) - \nabla_{\mathbf{x}} h_\mu(\mathbf{x}) \right\|_2^2 \Big| \mathcal{I} \right] \right]$$

$$\leq 2 \mathbb{E}_{\mathcal{I}} \left[ \mathbb{E}_{\mathbf{u}_i, i \in [q]} \left[ \left\| \widehat{\nabla}_{\mathbf{x}} h(\mathbf{x}) - \frac{1}{b} \sum_{j \in \mathcal{I}} \nabla_{\mathbf{x}} f_\mu(\mathbf{x}, \boldsymbol{\xi}_j) \right\|_2^2 + \left\| \frac{1}{b} \sum_{j \in \mathcal{I}} \nabla_{\mathbf{x}} f_\mu(\mathbf{x}, \boldsymbol{\xi}_j) - \nabla_{\mathbf{x}} h_\mu(\mathbf{x}) \right\|_2^2 \Big| \mathcal{I} \right] \right]. \tag{18}$$

Further, by definition, given $\mathcal{I}$, $\widehat{\nabla}_{\mathbf{x}} h(\mathbf{x})$ is the average of ZO gradient estimates under $q$ *i.i.d.* random directions, each of which has the mean $\frac{1}{b} \sum_{j \in \mathcal{I}} \nabla_{\mathbf{x}} f_\mu(\mathbf{x}, \boldsymbol{\xi}_j)$. Thus for the first term at the right-

hand-side (RHS) of the above inequality, we have

$$
\mathbb{E}_{\mathbf{u}_i, i\in[q]}\left[\left\|\widehat{\nabla}_{\mathbf{x}}h(\mathbf{x}) - \frac{1}{b}\sum_{j\in\mathcal{I}}\nabla_{\mathbf{x}}f_\mu(\mathbf{x}, \boldsymbol{\xi}_j)\right\|_2^2\bigg|\mathcal{I}\right] \le \frac{1}{q}\left(2d\left\|\frac{1}{b}\sum_{j\in\mathcal{I}}\nabla_{\mathbf{x}}f(\mathbf{x}, \boldsymbol{\xi}_j)\right\|^2 + \frac{\mu^2 L_h^2 d^2}{2}\right)
$$

$$
\le \frac{1}{q}\left(2d\eta^2 + \frac{\mu^2 L_h^2 d^2}{2}\right) \tag{19}
$$

where the first inequality is by the standard bound of the variance of zeroth order estimator and the second inequality is by the assumption that $\|\nabla_{\mathbf{x}}h(x; \boldsymbol{\xi})\|^2 \le \eta^2$ and thus $\|\frac{1}{b}\sum_{j\in\mathcal{I}}\nabla_{\mathbf{x}}f(\mathbf{x}, \boldsymbol{\xi}_j)\|^2 \le \eta^2$.In addition, we have

$$
\mathbb{E}_{\mathcal{I}}\left[\mathbb{E}_{\mathbf{u}_i, i\in[q]}\left[\left\|\frac{1}{b}\sum_{j\in\mathcal{I}}\nabla_{\mathbf{x}}f_\mu(\mathbf{x}, \boldsymbol{\xi}_j) - \nabla_{\mathbf{x}}h_\mu(\mathbf{x})\right\|_2^2\bigg|\mathcal{I}\right]\right]
$$

$$
=\mathbb{E}_{\mathcal{I}}\left[\left\|\frac{1}{b}\sum_{j\in\mathcal{I}}\nabla_{\mathbf{x}}f_\mu(\mathbf{x}, \boldsymbol{\xi}_j) - \nabla_{\mathbf{x}}h_\mu(\mathbf{x})\right\|_2^2\right]
$$

$$
=\frac{1}{b}\mathbb{E}_\xi\left[\|\nabla_{\mathbf{x}}f_\mu(\mathbf{x}, \boldsymbol{\xi}) - \nabla_{\mathbf{x}}h_\mu(\mathbf{x})\|_2^2\right] \le \frac{\eta^2}{b} \tag{20}
$$

where the second equality is because $\xi_j$ are i.i.d. draws from the same distribution as $\xi$ and $\mathbb{E}[\nabla_{\mathbf{x}}f_\mu(\mathbf{x}, \boldsymbol{\xi})] = \nabla_{\mathbf{x}}h_\mu(\mathbf{x})$, the last inequality is because $\|\nabla_{\mathbf{x}}f_\mu(\mathbf{x}, \boldsymbol{\xi})\|_2^2 \le \eta^2$ by assumption. Substituting (19) and (20) into (18) finishes the proof. □

### A.3 Convergence Analysis of ZO-Min-Max by Performing PGA

In this section, we will provide the details of the proofs. Before proceeding, we have the following illustration, which will be useful in the proof.

**The order of taking expectation:** Since iterates $\mathbf{x}^{(t)}, \mathbf{y}^{(t)}, \forall t$ are random variables, we need to define

$$
\mathcal{F}^{(t)} = \{\mathbf{x}^{(t)}, \mathbf{y}^{(t)}, \mathbf{x}^{(t-1)}, \mathbf{y}^{(t-1)}, \dots, \mathbf{x}^{(1)}, \mathbf{y}^{(1)}\} \tag{21}
$$

as the history of the iterates. Throughout the theoretical analysis, taking expectation means that we take expectation over random variable at the $t$th iteration conditioned on $\mathcal{F}^{(t-1)}$ and then take expectation over $\mathcal{F}^{(t-1)}$.

**Subproblem:** Also, it is worthy noting that performing (4) and (5) are equivalent to the following optimization problem:

$$
\mathbf{x}^{(t)} = \min_{\mathbf{x}\in\mathcal{X}}\left\langle\widehat{\nabla}_{\mathbf{x}}f(\mathbf{x}^{(t-1)}, \mathbf{y}^{(t-1)}), \mathbf{x} - \mathbf{x}^{(t-1)}\right\rangle + \frac{1}{2\alpha}\|\mathbf{x} - \mathbf{x}^{(t-1)}\|^2, \tag{22}
$$

$$
\mathbf{y}^{(t)} = \max_{\mathbf{y}\in\mathcal{Y}}\left\langle\widehat{\nabla}_{\mathbf{y}}f(\mathbf{x}^{(t)}, \mathbf{y}^{(t-1)}), \mathbf{y} - \mathbf{y}^{(t-1)}\right\rangle - \frac{1}{2\beta}\|\mathbf{y} - \mathbf{y}^{(t-1)}\|^2. \tag{23}
$$

When $f(\mathbf{x}, \mathbf{y})$ is white-box w.r.t. $\mathbf{y}$, (23) becomes

$$
\mathbf{y}^{(t)} = \max_{\mathbf{y}\in\mathcal{Y}}\left\langle\nabla_{\mathbf{y}}f(\mathbf{x}^{(t)}, \mathbf{y}^{(t-1)}), \mathbf{y} - \mathbf{y}^{(t-1)}\right\rangle - \frac{1}{2\beta}\|\mathbf{y} - \mathbf{y}^{(t-1)}\|^2. \tag{24}
$$

In the proof of ZO-Min-Max, we will use the optimality condition of these two problems to derive the descent lemmas.

**Relationship with smoothing function**  We denote by $f_{\mu,x}(\mathbf{x}, \mathbf{y})$ the smoothing version of $f$ w.r.t. $\mathbf{x}$ with parameter $\mu > 0$. The similar definition holds for $f_{\mu,y}(\mathbf{x}, \mathbf{y})$. By taking $f_{\mu,x}(\mathbf{x}, \mathbf{y})$ as an example, under **A2** $f$ and $f_{\mu,x}$ has the following relationship (Gao et al., 2014, Lemma 4.1):

$$|f_{\mu,\mathbf{x}}(\mathbf{x}, \mathbf{y}) - f(\mathbf{x}, \mathbf{y}))| \leq \frac{L_x \mu^2}{2} \quad \text{and} \quad \|\nabla_\mathbf{x} f_{\mu,\mathbf{x}}(\mathbf{x}, \mathbf{y}) - \nabla_\mathbf{x} f(\mathbf{x}, \mathbf{y})\|_2^2 \leq \frac{\mu^2 d^2 L_x^2}{4}, \quad (25)$$

$$|f_{\mu,\mathbf{y}}(\mathbf{x}, \mathbf{y}) - f(\mathbf{x}, \mathbf{y}))| \leq \frac{L_y \mu^2}{2} \quad \text{and} \quad \|\nabla_\mathbf{y} f_{\mu,\mathbf{y}}(\mathbf{x}, \mathbf{y}) - \nabla_\mathbf{y} f(\mathbf{x}, \mathbf{y})\|_2^2 \leq \frac{\mu^2 d^2 L_y^2}{4}. \quad (26)$$

First, we will show the descent lemma in minimization as follows.

### A.3.1  PROOF OF LEMMA 2

**Proof**: Since $f(\mathbf{x}, \mathbf{y})$ has $L_x$ Lipschtiz continuous gradients with respect to $\mathbf{x}$, we have

$$f_\mu(\mathbf{x}^{(t+1)}, \mathbf{y}^{(t)}) \leq f_\mu(\mathbf{x}^{(t)}, \mathbf{y}^{(t)}) + \langle \nabla_\mathbf{x} f_\mu(\mathbf{x}^{(t)}, \mathbf{y}^{(t)}), \mathbf{x}^{(t+1)} - \mathbf{x}^{(t)} \rangle + \frac{L_x}{2} \| \mathbf{x}^{(t+1)} - \mathbf{x}^{(t)} \|^2$$

$$= f_\mu(\mathbf{x}^{(t)}, \mathbf{y}^{(t)}) + \langle \widehat{\nabla}_\mathbf{x} f(\mathbf{x}^{(t)}, \mathbf{y}^{(t)}), \mathbf{x}^{(t+1)} - \mathbf{x}^{(t)} \rangle + \frac{L_x}{2} \| \mathbf{x}^{(t+1)} - \mathbf{x}^{(t)} \|^2$$

$$+ \langle \nabla_\mathbf{x} f_\mu(\mathbf{x}^{(t)}, \mathbf{y}^{(t)}) - \widehat{\nabla}_\mathbf{x} f(\mathbf{x}^{(t)}, \mathbf{y}^{(t)}), \mathbf{x}^{(t+1)} - \mathbf{x}^{(t)} \rangle. \quad (27)$$

Recall that

$$\mathbf{x}^{(t+1)} = \text{proj}_\mathcal{X}(\mathbf{x}^{(t)} - \alpha \widehat{\nabla}_\mathbf{x} f(\mathbf{x}^{(t)}, \mathbf{y}^{(t)})), \quad (28)$$

From the optimality condition of $\mathbf{x}$-subproblem (22), we have

$$\langle \widehat{\nabla}_\mathbf{x} f(\mathbf{x}^{(t)}, \mathbf{y}^{(t)}), \mathbf{x}^{(t+1)} - \mathbf{x}^{(t)} \rangle \leq -\frac{1}{\alpha} \| \mathbf{x}^{(t+1)} - \mathbf{x}^{(t)} \|^2. \quad (29)$$

Here we use the fact that the optimality condition of problem (22) at the solution $\mathbf{x}^{(t+1)}$ yields $\langle \widehat{\nabla}_\mathbf{x} f(\mathbf{x}^{(t)}, \mathbf{y}^{(t)}) + (\mathbf{x}^{(t+1)} - \mathbf{x}^{(t)})/\alpha, \mathbf{x}^{(t+1)} - \mathbf{x} \rangle \leq 0$ for any $\mathbf{x} \in \mathcal{X}$. By setting $\mathbf{x} = \mathbf{x}^{(t)}$, we obtain (29).

In addition, we define another iterate generated by $\nabla_\mathbf{x} f_\mu(\mathbf{x}^{(t)}, \mathbf{y}^{(t)})$

$$\widehat{\mathbf{x}}^{(t+1)} = \text{proj}_\mathcal{X}(\mathbf{x}^{(t)} - \alpha \nabla_\mathbf{x} f_\mu(\mathbf{x}^{(t)}, \mathbf{y}^{(t)})). \quad (30)$$

Then, we can have

$$\langle \nabla_\mathbf{x} f_\mu(\mathbf{x}^{(t)}, \mathbf{y}^{(t)}) - \widehat{\nabla}_\mathbf{x} f(\mathbf{x}^{(t)}, \mathbf{y}^{(t)}), \mathbf{x}^{(t+1)} - \mathbf{x}^{(t)} \rangle$$

$$= \langle \nabla_\mathbf{x} f_\mu(\mathbf{x}^{(t)}, \mathbf{y}^{(t)}) - \widehat{\nabla}_\mathbf{x} f(\mathbf{x}^{(t)}, \mathbf{y}^{(t)}), \mathbf{x}^{(t+1)} - \mathbf{x}^{(t)} - (\widehat{\mathbf{x}}^{(t+1)} - \mathbf{x}^{(t)}) \rangle$$

$$+ \langle \nabla_\mathbf{x} f_\mu(\mathbf{x}^{(t)}, \mathbf{y}^{(t)}) - \widehat{\nabla}_\mathbf{x} f(\mathbf{x}^{(t)}, \mathbf{y}^{(t)}), \widehat{\mathbf{x}}^{(t+1)} - \mathbf{x}^{(t)} \rangle. \quad (31)$$

Due to the fact that $\mathbb{E}_\mathbf{u}[\widehat{\nabla}_\mathbf{x} f(\mathbf{x}^{(t)}, \mathbf{y}^{(t)})] = \nabla_\mathbf{x} f_\mu(\mathbf{x}^{(t)}, \mathbf{y}^{(t)})$, we further have

$$\mathbb{E}_\mathbf{u}[\langle \nabla_\mathbf{x} f_\mu(\mathbf{x}^{(t)}, \mathbf{y}^{(t)}) - \widehat{\nabla}_\mathbf{x} f(\mathbf{x}^{(t)}, \mathbf{y}^{(t)}), \widehat{\mathbf{x}}^{(t+1)} - \mathbf{x}^{(t)} \rangle] = 0. \quad (32)$$

Finally, we also have

$$\langle \nabla_\mathbf{x} f_\mu(\mathbf{x}^{(t)}, \mathbf{y}^{(t)}) - \widehat{\nabla}_\mathbf{x} f(\mathbf{x}^{(t)}, \mathbf{y}^{(t)}), \mathbf{x}^{(t+1)} - \mathbf{x}^{(t)} - (\widehat{\mathbf{x}}^{(t+1)} - \mathbf{x}^{(t)}) \rangle$$

$$\leq \frac{\alpha}{2} \|\nabla_\mathbf{x} f_\mu(\mathbf{x}^{(t)}, \mathbf{y}^{(t)}) - \widehat{\nabla}_\mathbf{x} f(\mathbf{x}^{(t)}, \mathbf{y}^{(t)})\|^2 + \frac{1}{2\alpha} \| \mathbf{x}^{(t+1)} - \mathbf{x}^{(t)} - (\widehat{\mathbf{x}}^{(t+1)} - \mathbf{x}^{(t)}) \|^2$$

$$\leq \alpha \|\nabla_\mathbf{x} f_\mu(\mathbf{x}^{(t)}, \mathbf{y}^{(t)}) - \widehat{\nabla}_\mathbf{x} f(\mathbf{x}^{(t)}, \mathbf{y}^{(t)})\|^2 \quad (33)$$

where the first inequality is due to Young's inequality, the second inequality is due to non-expansiveness of the projection operator. Thus

$$\mathbb{E}_\mathbf{u}[\langle \nabla_\mathbf{x} f_\mu(\mathbf{x}^{(t)}, \mathbf{y}^{(t)}) - \widehat{\nabla}_\mathbf{x} f(\mathbf{x}^{(t)}, \mathbf{y}^{(t)}), \mathbf{x}^{(t+1)} - \mathbf{x}^{(t)} - (\widehat{\mathbf{x}}^{(t+1)} - \mathbf{x}^{(t)}) \rangle]$$

$$\leq \mathbb{E}_\mathbf{u}[\alpha \|\nabla_\mathbf{x} f_\mu(\mathbf{x}^{(t)}, \mathbf{y}^{(t)}) - \widehat{\nabla}_\mathbf{x} f(\mathbf{x}^{(t)}, \mathbf{y}^{(t)})\|^2] \leq \alpha \sigma_x^2 \quad (34)$$

where $\sigma_x^2 := \sigma^2(L_x, b, q, d)$ which was defined in (3).

Combining all above, we have

$$\mathbb{E}[f_\mu(\mathbf{x}^{(t+1)}, \mathbf{y}^{(t)})] \leq \mathbb{E}[f_\mu(\mathbf{x}^{(t)}, \mathbf{y}^{(t)})] - \left(\frac{1}{\alpha} - \frac{L_x}{2}\right) \|\mathbf{x}^{(t+1)} - \mathbf{x}^{(t)}\|^2 + \alpha\sigma^2, \qquad (35)$$

and we request $\alpha \leq 1/L_x$, which completes the proof.

Using $|f_{\mu,x}(\mathbf{x}, \mathbf{y}) - f(\mathbf{x}, \mathbf{y}))| \leq \frac{L_x\mu^2}{2}$, we can get

$$\mathbb{E}[f(\mathbf{x}^{(t+1)}, \mathbf{y}^{(t)})] - \frac{L_x\mu^2}{2} \leq \mathbb{E}[f_\mu(\mathbf{x}^{(t+1)}, \mathbf{y}^{(t)})] \leq \mathbb{E}[f(\mathbf{x}^{(t+1)}, \mathbf{y}^{(t)})] + \frac{L_x\mu^2}{2}, \qquad (36)$$

so we are able to obtain from (3)

$$\mathbb{E}[f(\mathbf{x}^{(t+1)}, \mathbf{y}^{(t)})] \leq \mathbb{E}[f(\mathbf{x}^{(t)}, \mathbf{y}^{(t)})] - \left(\frac{1}{\alpha} - \frac{L_x}{2}\right) \|\mathbf{x}^{(t+1)} - \mathbf{x}^{(t)}\|^2 + \alpha\sigma_x^2 + L_x\mu^2. \qquad (37)$$

$\square$

**Corollary 1.**

$$\mathbb{E}\left\langle \widehat{\nabla} f(\mathbf{x}^{(t)}, \mathbf{y}^{(t-1)}) - \nabla f_\mu(\mathbf{x}^{(t)}, \mathbf{y}^{(t-1)}), \mathbf{y}^{(t)} - \mathbf{y}^{(t-1)} \right\rangle \leq \beta\sigma_y^2 \qquad (38)$$

$\sigma_y^2 := \sigma^2(L_y, b, q, d)$ *which was defined in* (3).

**Proof:**

Define

$$\widetilde{\mathbf{y}}^{(t)} = \text{proj}_{\mathcal{Y}}(\mathbf{y}^{(t)} - \beta\nabla_{\mathbf{y}} f_\mu(\mathbf{x}^{(t)}, \mathbf{y}^{(t-1)})), \qquad (39)$$

we have

$$\langle \nabla_{\mathbf{y}} f_\mu(\mathbf{x}^{(t)}, \mathbf{y}^{(t-1)}) - \widehat{\nabla}_{\mathbf{x}} f(\mathbf{x}^{(t)}, \mathbf{y}^{(t-1)}), \mathbf{y}^{(t)} - \mathbf{y}^{(t-1)} \rangle$$
$$= \langle \nabla_{\mathbf{y}} f_\mu(\mathbf{x}^{(t)}, \mathbf{y}^{(t-1)}) - \widehat{\nabla}_{\mathbf{y}} f(\mathbf{x}^{(t)}, \mathbf{y}^{(t-1)}), \mathbf{y}^{(t)} - \mathbf{y}^{(t-1)} - (\widetilde{\mathbf{y}}^{(t)} - \mathbf{y}^{(t-1)}) \rangle$$
$$+ \langle \nabla_{\mathbf{y}} f_\mu(\mathbf{x}^{(t)}, \mathbf{y}^{(t-1)}) - \widehat{\nabla}_{\mathbf{y}} f(\mathbf{x}^{(t)}, \mathbf{y}^{(t-1)}), \widetilde{\mathbf{y}}^{(t)} - \mathbf{y}^{(t-1)} \rangle. \qquad (40)$$

Due to the fact that $\mathbb{E}_{\mathbf{u}}[\widehat{\nabla}_{\mathbf{y}} f(\mathbf{x}^{(t)}, \mathbf{y}^{(t-1)})] = \nabla_{\mathbf{y}} f_\mu(\mathbf{x}^{(t)}, \mathbf{y}^{(t-1)})$, we further have

$$\mathbb{E}_{\mathbf{u}}[\langle \nabla_{\mathbf{y}} f_\mu(\mathbf{x}^{(t)}, \mathbf{y}^{(t-1)}) - \widehat{\nabla}_{\mathbf{y}} f(\mathbf{x}^{(t)}, \mathbf{y}^{(t-1)}), \widetilde{\mathbf{y}}^{(t)} - \mathbf{y}^{(t-1)} \rangle] = 0. \qquad (41)$$

Finally, we also have

$$\mathbb{E}_{\mathbf{u}}[\langle \nabla_{\mathbf{y}} f_\mu(\mathbf{x}^{(t)}, \mathbf{y}^{(t-1)}) - \widehat{\nabla}_{\mathbf{y}} f(\mathbf{x}^{(t)}, \mathbf{y}^{(t-1)}), \mathbf{y}^{(t)} - \mathbf{y}^{(t-1)} - (\widetilde{\mathbf{y}}^{(t)} - \mathbf{y}^{(t-1)}) \rangle]$$
$$\leq \mathbb{E}_{\mathbf{u}}[\frac{\beta}{2} \|\langle \nabla_{\mathbf{y}} f_\mu(\mathbf{x}^{(t)}, \mathbf{y}^{(t-1)}) - \widehat{\nabla}_{\mathbf{y}} f(\mathbf{x}^{(t)}, \mathbf{y}^{(t-1)})\|^2 + \frac{1}{2\beta} \|\mathbf{y}^{(t)} - \mathbf{y}^{(t-1)} - (\widetilde{\mathbf{y}}^{(t)} - \mathbf{y}^{(t-1)})\|^2]$$
$$\leq \mathbb{E}_{\mathbf{u}}[\beta \|\nabla_{\mathbf{y}} f_\mu(\mathbf{x}^{(t)}, \mathbf{y}^{(t-1)}) - \widehat{\nabla}_{\mathbf{y}} f(\mathbf{x}^{(t)}, \mathbf{y}^{(t-1)})\|^2] \leq \beta\sigma_y^2 \qquad (42)$$

where $\sigma_y^2 := \sigma^2(L_y, b, q, d)$ which was defined in (3).

$\square$

Next, before showing the proof of Lemma 3, we need the following lemma to show the recurrence of the size of the successive difference between two iterations.

**Lemma 5.** *Under assumption 1, assume iterates* $\mathbf{x}^{(t)}, \mathbf{y}^{(t)}$ *generated by algorithm 1. When* $f(\mathbf{x}^{(t)}, \mathbf{y})$ *is white-box, we have*

$$\frac{2}{\beta^2\gamma} \mathbb{E}\|\mathbf{y}^{(t+1)} - \mathbf{y}^{(t)}\|^2 - \frac{2}{\beta^2\gamma} \mathbb{E}\|\mathbf{y}^{(t)} - \mathbf{y}^{(t-1)}\|^2 \leq \frac{2L_x^2}{\beta\gamma^2} \mathbb{E}\|\mathbf{x}^{(t+1)} - \mathbf{x}^{(t)}\|^2$$

$$+ \frac{2}{\beta} \mathbb{E}\|\mathbf{y}^{(t+1)} - \mathbf{y}^{(t)}\|^2 - \left(\frac{4}{\beta} - \frac{2L_y^2}{\gamma}\right) \mathbb{E}\|\mathbf{y}^{(t)} - \mathbf{y}^{(t-1)}\|^2. \qquad (43)$$

**Proof**: from the optimality condition of $\mathbf{y}$-subproblem (24) at iteration $t$ and $t-1$, we have the following two inequalities:

$$-\langle \nabla_{\mathbf{y}} f(\mathbf{x}^{(t+1)}, \mathbf{y}^{(t)}) - \frac{1}{\beta}(\mathbf{y}^{(t+1)} - \mathbf{y}^{(t)}), \mathbf{y}^{(t+1)} - \mathbf{y}^{(t)}\rangle \leq 0, \tag{44}$$

$$\langle \nabla_{\mathbf{y}} f(\mathbf{x}^{(t)}, \mathbf{y}^{(t-1)}) - \frac{1}{\beta}(\mathbf{y}^{(t)} - \mathbf{y}^{(t-1)}), \mathbf{y}^{(t+1)} - \mathbf{y}^{(t)}\rangle \leq 0. \tag{45}$$

Adding the above inequalities, we can get

$$\frac{1}{\beta}\langle \mathbf{v}^{(t+1)}, \mathbf{y}^{(t+1)} - \mathbf{y}^{(t)}\rangle \leq \left\langle \nabla_{\mathbf{y}} f(\mathbf{x}^{(t+1)}, \mathbf{y}^{(t)}) - \nabla_{\mathbf{y}} f(\mathbf{x}^{(t)}, \mathbf{y}^{(t)}), \mathbf{y}^{(t+1)} - \mathbf{y}^{(t)}\right\rangle$$
$$+ \left\langle \nabla_{\mathbf{y}} f(\mathbf{x}^{(t)}, \mathbf{y}^{(t)}) - \nabla_{\mathbf{y}} f(\mathbf{x}^{(t)}, \mathbf{y}^{(t-1)}), \mathbf{y}^{(t+1)} - \mathbf{y}^{(t)}\right\rangle \tag{46}$$

where $\mathbf{v}^{(t+1)} = \mathbf{y}^{(t+1)} - \mathbf{y}^{(t)} - (\mathbf{y}^{(t)} - \mathbf{y}^{(t-1)})$.

According to the quadrilateral indentity, we know

$$\left\langle \mathbf{v}^{(t+1)}, \mathbf{y}^{(t+1)} - \mathbf{y}^{(t)}\right\rangle = \frac{1}{2}\left(\|\mathbf{y}^{(t+1)} - \mathbf{y}^{(t)}\|^2 + \|\mathbf{v}^{(t+1)}\|^2 - \|\mathbf{y}^{(t)} - \mathbf{y}^{(t-1)}\|^2\right). \tag{47}$$

Based on the definition of $\mathbf{v}^{(t+1)}$, we substituting (47) into (46), which gives

$$\frac{1}{2\beta}\|\mathbf{y}^{(t+1)} - \mathbf{y}^{(t)}\|^2 \leq \frac{1}{2\beta}\|\mathbf{y}^{(t)} - \mathbf{y}^{(t-1)}\|^2 - \frac{1}{2\beta}\|\mathbf{v}^{(t+1)}\|^2$$
$$+ \left\langle \nabla_{\mathbf{y}} f(\mathbf{x}^{(t+1)}, \mathbf{y}^{(t)}) - \nabla_{\mathbf{y}} f(\mathbf{x}^{(t)}, \mathbf{y}^{(t)}), \mathbf{y}^{(t+1)} - \mathbf{y}^{(t)}\right\rangle$$
$$+ \left\langle \nabla_{\mathbf{y}} f(\mathbf{x}^{(t)}, \mathbf{y}^{(t)}) - \nabla_{\mathbf{y}} f(\mathbf{x}^{(t)}, \mathbf{y}^{(t-1)}), \mathbf{y}^{(t+1)} - \mathbf{y}^{(t)}\right\rangle \tag{48}$$
$$\overset{(a)}{\leq} \frac{1}{2\beta}\|\mathbf{y}^{(t)} - \mathbf{y}^{(t-1)}\|^2 + \left\langle \nabla_{\mathbf{y}} f(\mathbf{x}^{(t+1)}, \mathbf{y}^{(t)}) - \nabla_{\mathbf{y}} f(\mathbf{x}^{(t)}, \mathbf{y}^{(t)}), \mathbf{y}^{(t+1)} - \mathbf{y}^{(t)}\right\rangle$$
$$+ \frac{\beta L_y^2}{2}\|\mathbf{y}^{(t)} - \mathbf{y}^{(t-1)}\|^2 - \gamma\|\mathbf{y}^{(t)} - \mathbf{y}^{(t-1)}\|^2$$
$$\overset{(b)}{\leq} \frac{1}{2\beta}\|\mathbf{y}^{(t)} - \mathbf{y}^{(t-1)}\|^2 + \frac{\gamma}{2}\|\mathbf{y}^{(t+1)} - \mathbf{y}^{(t)}\|^2$$
$$+ \frac{L_x^2}{2\gamma}\|\mathbf{x}^{(t+1)} - \mathbf{x}^{(t)}\|^2 - (\gamma - \frac{\beta L_y^2}{2})\|\mathbf{y}^{(t)} - \mathbf{y}^{(t-1)}\|^2 \tag{49}$$

where in $(a)$ we use the strong concavity of function $f(\mathbf{x}, \mathbf{y})$ in $\mathbf{y}$ (with parameter $\gamma > 0$) and Young's inequality, i.e.,

$$\langle \nabla_{\mathbf{y}} f(\mathbf{x}^{(t)}, \mathbf{y}^{(t)}) - \nabla_{\mathbf{y}} f(\mathbf{x}^{(t)}, \mathbf{y}^{(t-1)}), \mathbf{y}^{(t+1)} - \mathbf{y}^{(t)}\rangle$$
$$= \langle \nabla_{\mathbf{y}} f(\mathbf{x}^{(t)}, \mathbf{y}^{(t)}) - \nabla_{\mathbf{y}} f(\mathbf{x}^{(t)}, \mathbf{y}^{(t-1)}), \mathbf{v}^{(t+1)} + \mathbf{y}^{(t)} - \mathbf{y}^{(t-1)}\rangle$$
$$\leq \frac{\beta L_y^2}{2}\|\mathbf{y}^{(t)} - \mathbf{y}^{(t-1)}\|^2 + \frac{1}{2\beta}\|\mathbf{v}^{(t+1)}\|^2 - \gamma\|\mathbf{y}^{(t)} - \mathbf{y}^{(t-1)}\|^2 \tag{50}$$

and in $(b)$ we apply the Young's inequality, i.e.,

$$\left\langle \nabla_{\mathbf{y}} f(\mathbf{x}^{(t+1)}, \mathbf{y}^{(t)}) - \nabla_{\mathbf{y}} f(\mathbf{x}^{(t)}, \mathbf{y}^{(t)}), \mathbf{y}^{(t+1)} - \mathbf{y}^{(t)}\right\rangle \leq \frac{L_x^2}{2\gamma}\|\mathbf{x}^{(t+1)} - \mathbf{x}^{(t)}\|^2 + \frac{\gamma}{2}\|\mathbf{y}^{(t+1)} - \mathbf{y}^{(t)}\|^2. \tag{51}$$

Therefore, we have

$$\frac{1}{2\beta}\|\mathbf{y}^{(t+1)} - \mathbf{y}^{(t)}\|^2 \leq \frac{1}{2\beta}\|\mathbf{y}^{(t)} - \mathbf{y}^{(t-1)}\|^2 + \frac{L_x^2}{2\gamma}\|\mathbf{x}^{(t+1)} - \mathbf{x}^{(t)}\|^2$$
$$+ \frac{\gamma}{2}\|\mathbf{y}^{(t+1)} - \mathbf{y}^{(t)}\|^2 - \left(\gamma - \frac{\beta L_y^2}{2}\right)\|\mathbf{y}^{(t)} - \mathbf{y}^{(t-1)}\|^2, \tag{52}$$

which implies

$$\frac{2}{\beta^2\gamma}\|\mathbf{y}^{(t+1)}-\mathbf{y}^{(t)}\|^2 \leq \frac{2}{\beta^2\gamma}\|\mathbf{y}^{(t)}-\mathbf{y}^{(t-1)}\|^2 + \frac{2L_x^2}{\beta\gamma^2}\|\mathbf{x}^{(t+1)}-\mathbf{x}^{(t)}\|^2$$

$$+\frac{2}{\beta}\|\mathbf{y}^{(t+1)}-\mathbf{y}^{(t)}\|^2 - \left(\frac{4}{\beta}-\frac{2L_y^2}{\gamma}\right)\|\mathbf{y}^{(t)}-\mathbf{y}^{(t-1)}\|^2. \qquad (53)$$

By taking the expectation on both sides of (53), we can get the results of Lemma 5. □

Lemma 5 basically gives the recursion of $\|\Delta_y^{(t)}\|^2$. It can be observed that term $(4/\beta - 2L_y^2/\gamma)\|\Delta_y^{(t)}\|$ provides the descent of the recursion when $\beta$ is small enough, which will take an important role in the proof of Lemma 3 when we quantify the descent in maximization.

Then, we can quantify the descent of the objective value by the following descent lemma.

### A.3.2 PROOF OF LEMMA 3

**Proof**: let $f'(\mathbf{x}^{(t+1)},\mathbf{y}^{(t+1)}) = f(\mathbf{x}^{(t+1)},\mathbf{y}^{(t+1)}) - 1(\mathbf{y}^{(t+1)})$ and $1(\mathbf{y})$ denote the indicator function with respect to the constraint of $\mathbf{y}$. From the optimality condition of sub-problem $\mathbf{y}$ in (23), we have

$$\nabla_\mathbf{y} f(\mathbf{x}^{(t+1)},\mathbf{y}^{(t)}) - \frac{1}{\beta}(\mathbf{y}^{(t+1)}-\mathbf{y}^{(t)}) - \xi^{(t+1)} = 0 \qquad (54)$$

where $\xi^{(t)}$ denote the subgradient of $1(\mathbf{y}^{(t)})$. Since function $f'(\mathbf{x},\mathbf{y})$ is concave with respect to $\mathbf{y}$, we have

$$f'(\mathbf{x}^{(t+1)},\mathbf{y}^{(t+1)}) - f'(\mathbf{x}^{(t+1)},\mathbf{y}^{(t)}) \leq \langle\nabla_\mathbf{y} f(\mathbf{x}^{(t+1)},\mathbf{y}^{(t)}),\mathbf{y}^{(t+1)}-\mathbf{y}^{(t)}\rangle - \langle\xi^{(t)},\mathbf{y}^{(t+1)}-\mathbf{y}^{(t)}\rangle$$

$$\overset{(a)}{=}\frac{1}{\beta}\|\mathbf{y}^{(t+1)}-\mathbf{y}^{(t)}\|^2 - \langle\xi^{(t)}-\xi^{(t+1)},\mathbf{y}^{(t+1)}-\mathbf{y}^{(t)}\rangle$$

$$=\frac{1}{\beta}\|\mathbf{y}^{(t+1)}-\mathbf{y}^{(t)}\|^2 + \left\langle\nabla_\mathbf{y} f(\mathbf{x}^{(t+1)},\mathbf{y}^{(t)}) - \nabla_\mathbf{y} f(\mathbf{x}^{(t)},\mathbf{y}^{(t-1)}),\mathbf{y}^{(t+1)}-\mathbf{y}^{(t)}\right\rangle$$

$$-\frac{1}{\beta}\left\langle\mathbf{v}^{(t+1)},\mathbf{y}^{(t+1)}-\mathbf{y}^{(t)}\right\rangle \qquad (55)$$

where in $(a)$ we use $\xi^{(t+1)} = \nabla_\mathbf{y} f(\mathbf{x}^{(t+1)},\mathbf{y}^{(t)}) - \frac{1}{\beta}(\mathbf{y}^{(t+1)}-\mathbf{y}^{(t)})$. The last two terms of (55) is the same as the RHS of (46). We can apply the similar steps from (48) to (49). To be more specific, the derivations are shown as follows: First, we know

$$f'(\mathbf{x}^{(t+1)},\mathbf{y}^{(t+1)}) - f'(\mathbf{x}^{(t+1)},\mathbf{y}^{(t)}) \leq \frac{1}{\beta}\|\mathbf{y}^{(t+1)}-\mathbf{y}^{(t)}\|^2$$

$$+\left\langle\nabla_\mathbf{y} f(\mathbf{x}^{(t+1)},\mathbf{y}^{(t)}) - \nabla_\mathbf{y} f(\mathbf{x}^{(t)},\mathbf{y}^{(t-1)}),\mathbf{y}^{(t+1)}-\mathbf{y}^{(t)}\right\rangle - \frac{1}{\beta}\left\langle\mathbf{v}^{(t+1)},\mathbf{y}^{(t+1)}-\mathbf{y}^{(t)}\right\rangle. \quad (56)$$

Then, we move term $1/\beta\langle\mathbf{v}^{(t+1)},\mathbf{y}^{(t+1)}-\mathbf{y}^{(t)}\rangle$ to RHS of (55) and have

$$f(\mathbf{x}^{(t+1)},\mathbf{y}^{(t+1)}) - f(\mathbf{x}^{(t+1)},\mathbf{y}^{(t)})$$

$$\leq\frac{1}{2\beta}\|\mathbf{y}^{(t+1)}-\mathbf{y}^{(t)}\|^2 + \frac{1}{2\beta}\|\mathbf{y}^{(t)}-\mathbf{y}^{(t-1)}\|^2 - \frac{1}{2\beta}\|\mathbf{v}^{(t+1)}\|^2$$

$$+\left\langle\nabla_\mathbf{y} f(\mathbf{x}^{(t+1)},\mathbf{y}^{(t)}) - \nabla_\mathbf{y} f(\mathbf{x}^{(t)},\mathbf{y}^{(t)}),\mathbf{y}^{(t+1)}-\mathbf{y}^{(t)}\right\rangle$$

$$+\left\langle\nabla_\mathbf{y} f(\mathbf{x}^{(t)},\mathbf{y}^{(t)}) - \nabla_\mathbf{y} f(\mathbf{x}^{(t)},\mathbf{y}^{(t-1)}),\mathbf{y}^{(t+1)}-\mathbf{y}^{(t)}\right\rangle$$

$$\leq\frac{1}{2\beta}\|\mathbf{y}^{(t+1)}-\mathbf{y}^{(t)}\|^2 + \left\langle\nabla_\mathbf{y} f(\mathbf{x}^{(t+1)},\mathbf{y}^{(t)}) - \nabla_\mathbf{y} f(\mathbf{x}^{(t)},\mathbf{y}^{(t)}),\mathbf{y}^{(t+1)}-\mathbf{y}^{(t)}\right\rangle$$

$$+\frac{\beta L_y^2}{2}\|\mathbf{y}^{(t)}-\mathbf{y}^{(t-1)}\|^2 - \gamma\|\mathbf{y}^{(t)}-\mathbf{y}^{(t-1)}\|^2$$

$$\overset{(a)}{\leq}\frac{1}{\beta}\|\mathbf{y}^{(t+1)}-\mathbf{y}^{(t)}\|^2 + \frac{1}{2\beta}\|\mathbf{y}^{(t)}-\mathbf{y}^{(t-1)}\|^2$$

$$+\frac{\beta L_x^2}{2}\|\mathbf{x}^{(t+1)}-\mathbf{x}^{(t)}\|^2 - (\gamma-\frac{\beta L_y^2}{2})\|\mathbf{y}^{(t)}-\mathbf{y}^{(t-1)}\|^2 \qquad (57)$$

where in $(a)$ we use

$$\langle \nabla_{\mathbf{y}} f(\mathbf{x}^{(t+1)}, \mathbf{y}^{(t)}) - \nabla_{\mathbf{y}} f(\mathbf{x}^{(t)}, \mathbf{y}^{(t)}) \rangle \leq \frac{\beta L_x^2}{2} \| \mathbf{x}^{(t+1)} - \mathbf{x}^{(t)} \|^2 + \frac{1}{2\beta} \| \mathbf{y}^{(t+1)} - \mathbf{y}^{(t)} \|^2 \quad (58)$$

which is different from (51); also $\mathbf{y}^{(t)}, \mathbf{y}^{(t+1)} \in \mathcal{Y}$ so have $f'(\mathbf{x}^{(t+1)}, \mathbf{y}^{(t+1)}) = f(\mathbf{x}^{(t+1)}, \mathbf{y}^{(t+1)})$ and $f'(\mathbf{x}^{(t+1)}, \mathbf{y}^{(t)}) = f(\mathbf{x}^{(t+1)}, \mathbf{y}^{(t)})$.

Combing (53), we have

$$f(\mathbf{x}^{(t+1)}, \mathbf{y}^{(t+1)}) + \left( \frac{2}{\beta^2 \gamma} + \frac{1}{2\beta} \right) \| \mathbf{y}^{(t+1)} - \mathbf{y}^{(t)} \|^2 - 4 \left( \frac{1}{\beta} - \frac{L_y^2}{2\gamma} \right) \| \mathbf{y}^{(t+1)} - \mathbf{y}^{(t)} \|^2$$

$$\leq f(\mathbf{x}^{(t+1)}, \mathbf{y}^{(t)}) + \left( \frac{2}{\beta^2 \gamma} + \frac{1}{2\beta} \right) \| \mathbf{y}^{(t)} - \mathbf{y}^{(t-1)} \|^2 - 4 \left( \frac{1}{\beta} - \frac{L_y^2}{2\gamma} \right) \| \mathbf{y}^{(t)} - \mathbf{y}^{(t-1)} \|^2$$

$$- \left( \frac{1}{2\beta} - \frac{2L_y^2}{\gamma} \right) \| \mathbf{y}^{(t+1)} - \mathbf{y}^{(t)} \|^2 + \left( \frac{2L_x^2}{\gamma^2 \beta} + \frac{\beta L_x^2}{2} \right) \| \mathbf{x}^{(t+1)} - \mathbf{x}^{(t)} \|^2. \quad (59)$$

By taking the expectation on both sides of (53), we can get the results of Lemma 3. $\square$

Next, we use the following lemma to show the descent of the objective value after solving $\mathbf{x}$-subproblem by (4).

### A.3.3 PROOF OF THEOREM 1

**Proof**:

From Lemma 3, we know

$$\mathbb{E}[f(\mathbf{x}^{(t+1)}, \mathbf{y}^{(t+1)})] + \left( \frac{2}{\beta^2 \gamma} + \frac{1}{2\beta} \right) \mathbb{E}[\| \mathbf{y}^{(t+1)} - \mathbf{y}^{(t)} \|^2]$$

$$- 4 \left( \frac{1}{\beta} - \frac{L_y^2}{2\gamma} \right) \mathbb{E}[\| \mathbf{y}^{(t+1)} - \mathbf{y}^{(t)} \|^2] \leq \mathbb{E}[f(\mathbf{x}^{(t+1)}, \mathbf{y}^{(t)})]$$

$$+ \left( \frac{2}{\beta^2 \gamma} + \frac{1}{2\beta} \right) \mathbb{E}[\| \mathbf{y}^{(t)} - \mathbf{y}^{(t-1)} \|^2] - 4 \left( \frac{1}{\beta} - \frac{L_y^2}{2\gamma} \right) \mathbb{E}[\| \mathbf{y}^{(t)} - \mathbf{y}^{(t-1)} \|^2]$$

$$- \left( \frac{1}{2\beta} - \frac{2L_y^2}{\gamma} \right) \mathbb{E}[\| \mathbf{y}^{(t+1)} - \mathbf{y}^{(t)} \|^2] + \left( \frac{2L_x^2}{\gamma^2 \beta} + \frac{\beta L_x^2}{2} \right) \mathbb{E}[\| \mathbf{x}^{(t+1)} - \mathbf{x}^{(t)} \|^2]. \quad (60)$$

Combining Lemma 2, we have

$$\mathbb{E}[f(\mathbf{x}^{(t+1)}, \mathbf{y}^{(t+1)})] + \left( \frac{2}{\beta^2 \gamma} + \frac{1}{2\beta} \right) \mathbb{E}\left[ \| \mathbf{y}^{(t+1)} - \mathbf{y}^{(t)} \|^2 \right]$$

$$- 4 \left( \frac{1}{\beta} - \frac{L_y^2}{2\gamma} \right) \mathbb{E}\left[ \| \mathbf{y}^{(t+1)} - \mathbf{y}^{(t)} \|^2 \right] \leq \mathbb{E}[f(\mathbf{x}^{(t)}, \mathbf{y}^{(t)})] + \left( \frac{2}{\beta^2 \gamma} + \frac{1}{2\beta} \right) \mathbb{E}\left[ \| \mathbf{y}^{(t)} - \mathbf{y}^{(t-1)} \|^2 \right]$$

$$- 4 \left( \frac{1}{\beta} - \frac{L_y^2}{2\gamma} \right) \mathbb{E}\left[ \| \mathbf{y}^{(t)} - \mathbf{y}^{(t-1)} \|^2 \right] - \underbrace{\left( \frac{1}{2\beta} - \frac{2L_y^2}{\gamma} \right)}_{c_1} \mathbb{E}\left[ \| \mathbf{y}^{(t+1)} - \mathbf{y}^{(t)} \|^2 \right]$$

$$- \underbrace{\left( \frac{1}{\alpha} - \left( \frac{L_x}{2} + \frac{2L_x^2}{\gamma^2 \beta} + \frac{\beta L_x^2}{2} \right) \right)}_{c_2} \mathbb{E}\left[ \| \mathbf{x}^{(t+1)} - \mathbf{x}^{(t)} \|^2 \right] + \alpha \sigma_x^2 + L_x \mu^2. \quad (61)$$

If

$$\beta < \frac{\gamma}{4L_y^2} \quad \text{and} \quad \alpha < \frac{1}{\frac{L_x}{2} + \frac{2L_x^2}{\gamma^2 \beta} + \frac{\beta L_x^2}{2}}, \quad (62)$$

then we have that there exist positive constants $c_1$ and $c_2$ such that

$$\mathcal{P}(\mathbf{x}^{(t+1)}, \mathbf{y}^{(t+1)}, \Delta_{\mathbf{y}}^{(t+1)}) - \mathcal{P}(\mathbf{x}^{(t)}, \mathbf{y}^{(t)}, \Delta_{\mathbf{y}}^{(t)})$$

$$\leq - c_1 \mathbb{E}\left[\|\mathbf{y}^{(t+1)} - \mathbf{y}^{(t)}\|^2\right] - c_2 \mathbb{E}\left[\|\mathbf{x}^{(t+1)} - \mathbf{x}^{(t)}\|^2\right] + \alpha\sigma_x^2 + L_x\mu^2$$

$$\leq - \zeta\left(\mathbb{E}\left[\|\mathbf{y}^{(t+1)} - \mathbf{y}^{(t)}\|^2\right] + \mathbb{E}\left[\|\mathbf{x}^{(t+1)} - \mathbf{x}^{(t)}\|^2\right]\right) + \alpha\sigma_x^2 + L_x\mu^2 \qquad (63)$$

where $\zeta = \min\{c_1, c_2\}$.

From (6), we can have

$$\|\mathcal{G}(\mathbf{x}^{(t)}, \mathbf{y}^{(t)})\|$$

$$\leq \frac{1}{\alpha}\|\mathbf{x}^{(t+1)} - \mathbf{x}^{(t)}\| + \frac{1}{\alpha}\|\mathbf{x}^{(t+1)} - \mathrm{proj}_{\mathcal{X}}(\mathbf{x}^{(t)} - \alpha\nabla_{\mathbf{x}}f(\mathbf{x}^{(t)}, \mathbf{y}^{(t)}))\| + \frac{1}{\beta}\|\mathbf{y}^{(t+1)} - \mathbf{y}^{(t)}\|$$

$$+ \frac{1}{\beta}\|\mathbf{y}^{(t+1)} - \mathrm{proj}_{\mathcal{Y}}(\mathbf{y}^{(t)} + \beta\nabla_{\mathbf{y}}f(\mathbf{x}^{(t)}, \mathbf{y}^{(t)}))\|$$

$$\overset{(a)}{\leq} \frac{1}{\alpha}\|\mathbf{x}^{(t+1)} - \mathbf{x}^{(t)}\|$$

$$+ \frac{1}{\alpha}\|\mathrm{proj}_{\mathcal{X}}(\mathbf{x}^{(t+1)} - \alpha(\nabla_{\mathbf{x}}f(\mathbf{x}^{(t)}, \mathbf{y}^{(t)}) + \frac{1}{\alpha}(\mathbf{x}^{(t+1)} - \mathbf{x}^{(t)})) - \mathrm{proj}_{\mathcal{X}}(\mathbf{x}^{(t)} - \alpha\nabla_{\mathbf{x}}f(\mathbf{x}^{(t)}, \mathbf{y}^{(t)}))\|$$

$$+ \frac{1}{\beta}\|\mathbf{y}^{(t+1)} - \mathbf{y}^{(t)}\|$$

$$+ \frac{1}{\beta}\|\mathrm{proj}_{\mathcal{Y}}(\mathbf{y}^{(t+1)} + \beta(\nabla_{\mathbf{y}}f(\mathbf{x}^{(t+1)}, \mathbf{y}^{(t)}) - \frac{1}{\beta}(\mathbf{y}^{(t+1)} - \mathbf{y}^{(t)})) - \mathrm{proj}_{\mathcal{Y}}(\mathbf{y}^{(t)} + \beta\nabla_{\mathbf{y}}f(\mathbf{x}^{(t)}, \mathbf{y}^{(t)}))\|$$

$$\overset{(b)}{\leq} \frac{3}{\alpha}\|\mathbf{x}^{(t+1)} - \mathbf{x}^{(t)}\| + \|\nabla_{\mathbf{y}}f(\mathbf{x}^{(t+1)}, \mathbf{y}^{(t)}) - \nabla_{\mathbf{y}}f(\mathbf{x}^{(t)}, \mathbf{y}^{(t)})\| + \frac{3}{\beta}\|\mathbf{y}^{(t+1)} - \mathbf{y}^{(t)}\|$$

$$\overset{(c)}{\leq} \left(\frac{3}{\alpha} + L_x\right)\|\mathbf{x}^{(t+1)} - \mathbf{x}^{(t)}\| + \frac{3}{\beta}\|\mathbf{y}^{(t+1)} - \mathbf{y}^{(t)}\|$$

where in $(a)$ we use $\mathbf{x}^{(t+1)} = \mathrm{proj}_{\mathcal{X}}(\mathbf{x}^{(t+1)} - \alpha\nabla f(\mathbf{x}^{(t+1)}, \mathbf{y}^{(t)}) - (\mathbf{x}^{(t+1)} - \mathbf{x}^{(t)}))$; in $(b)$ we use nonexpansiveness of the projection operator; in $(c)$ we apply the Lipschitz continuous of function $f(\mathbf{x}, \mathbf{y})$ with respect to $\mathbf{x}$ and $\mathbf{y}$ under assumption **A2**. Therefore, we can know that there exist a constant $c = \max\{L_x + \frac{3}{\alpha}, \frac{3}{\beta}\}$ such that

$$\|\mathcal{G}(\mathbf{x}^{(t)}, \mathbf{y}^{(t)})\|^2 \leq c\left(\|\mathbf{x}^{(t+1)} - \mathbf{x}^{(t)}\|^2 + \|\mathbf{y}^{(t+1)} - \mathbf{y}^{(t)}\|^2\right). \qquad (64)$$

After applying the telescope sum on (63) and taking expectation over (64), we have

$$\frac{1}{T}\sum_{t=1}^{T}\mathbb{E}\|\mathcal{G}(\mathbf{x}^{(t)}, \mathbf{y}^{(t)})\|^2 \leq \frac{c}{\zeta}\left(\frac{\mathcal{P}_1 - \mathcal{P}_{T+1}}{T} + \alpha\sigma_x^2 + L_x\mu^2\right). \qquad (65)$$

Recall from **A1** that $f \geq f^*$ and $\mathcal{Y}$ is bounded with diameter $R$, therefore, $\mathcal{P}_t$ given by (8) yields

$$\mathcal{P}_t \geq f^* + \left(\frac{\min\{4 + 4\beta^2 L_y^2 - 7\beta\gamma, 0\}}{2\beta^2\gamma}\right) R^2, \quad \forall t. \qquad (66)$$

And let $(\mathbf{x}^{(r)}, \mathbf{y}^{(r)})$ be uniformly and randomly picked from $\{(\mathbf{x}^{(t)}, \mathbf{y}^{(t)})\}_{t=1}^T$, based on (65) and (66), we obtain

$$\mathbb{E}_r[\mathbb{E}\|\mathcal{G}(\mathbf{x}^{(r)}, \mathbf{y}^{(r)})\|^2] = \frac{1}{T}\sum_{t=1}^{T}\mathbb{E}\|\mathcal{G}(\mathbf{x}^{(t)}, \mathbf{y}^{(t)})\|^2 \leq \frac{c}{\zeta}\left(\frac{\mathcal{P}_1 - f^* - \nu R^2}{T} + \alpha\sigma_x^2 + L_x\mu^2\right),$$

$$(67)$$

where recall that $\zeta = \min\{c_1, c_2\}$, $c = \max\{L_x + \frac{3}{\alpha}, \frac{3}{\beta}\}$ and $\nu = \frac{\min\{4 + 4\beta^2 L_y^2 - 7\beta\gamma, 0\}}{2\beta^2\gamma}$.

The proof is now complete. $\square$

A.4 CONVERGENCE ANALYSIS OF ZO-MIN-MAX BY PERFORMING ZO-PGA

Before showing the proof of Lemma 4, we first give the following lemma regarding to recursion of the difference between two successive iterates of variable $\mathbf{y}$.

**Lemma 6.** *Under assumption 1, assume iterates* $\mathbf{x}^{(t)}, \mathbf{y}^{(t)}$ *generated by algorithm 1. When function* $f(\mathbf{x}^{(t)}, \mathbf{y})$ *is black-box, we have*

$$
\begin{aligned}
\frac{2}{\beta^2\gamma}\mathbb{E}\|\mathbf{y}^{(t+1)} - \mathbf{y}^{(t)}\|^2 \leq & \frac{2}{\beta^2\gamma}\mathbb{E}\|\mathbf{y}^{(t)} - \mathbf{y}^{(t-1)}\|^2 + \frac{2}{\beta}\mathbb{E}\|\mathbf{y}^{(t+1)} - \mathbf{y}^{(t)}\|^2 \\
& + \frac{6L_y^2}{\beta\gamma^2}\mathbb{E}\|\mathbf{x}^{(t+1)} - \mathbf{x}^{(t)}\|^2 - \left(\frac{4}{\beta} - \frac{6L_y^2 + 4}{\gamma}\right)\mathbb{E}\|\mathbf{y}^{(t)} - \mathbf{y}^{(t-1)}\|^2 \\
& + \frac{4\sigma_y^2}{\beta\gamma}\left(\frac{3}{\gamma} + 4\beta\right) + \frac{\mu^2 d^2 L_y^2}{\beta^2\gamma}.
\end{aligned}
\tag{68}
$$

From the optimality condition of $\mathbf{y}$-subproblem in (23) at iteration $t$ and $t-1$, we have

$$
-\left\langle \widehat{\nabla}_{\mathbf{y}}f(\mathbf{x}^{(t+1)}, \mathbf{y}^{(t)}) - \frac{1}{\beta}(\mathbf{y}^{(t+1)} - \mathbf{y}^{(t)}), \mathbf{y}^{(t+1)} - \mathbf{y}^{(t)}\right\rangle \leq 0,
\tag{69}
$$

$$
\left\langle \widehat{\nabla}_{\mathbf{y}}f(\mathbf{x}^{(t)}, \mathbf{y}^{(t-1)}) - \frac{1}{\beta}(\mathbf{y}^{(t)} - \mathbf{y}^{(t-1)}), \mathbf{y}^{(t+1)} - \mathbf{y}^{(t)}\right\rangle \leq 0.
\tag{70}
$$

Adding the above inequalities and applying the definition of $\mathbf{v}^{(t+1)}$, we can get

$$
\begin{aligned}
\frac{1}{\beta}\langle \mathbf{v}^{(t+1)}, \mathbf{y}^{(t+1)} - \mathbf{y}^{(t)}\rangle \leq & \underbrace{\left\langle \widehat{\nabla}_{\mathbf{y}}f(\mathbf{x}^{(t+1)}, \mathbf{y}^{(t)}) - \widehat{\nabla}_{\mathbf{y}}f(\mathbf{x}^{(t)}, \mathbf{y}^{(t)}), \mathbf{y}^{(t+1)} - \mathbf{y}^{(t)}\right\rangle}_{\mathbf{I}} \\
& + \underbrace{\left\langle \widehat{\nabla}_{\mathbf{y}}f(\mathbf{x}^{(t)}, \mathbf{y}^{(t)}) - \widehat{\nabla}_{\mathbf{y}}f(\mathbf{x}^{(t)}, \mathbf{y}^{(t-1)}), \mathbf{y}^{(t+1)} - \mathbf{y}^{(t)}\right\rangle}_{\mathbf{II}}.
\end{aligned}
\tag{71}
$$

Next, we will bound $\mathbb{E}[\mathbf{I}]$ and $\mathbb{E}[\mathbf{II}]$ separably as follows.

First, we give an upper bound of $\mathbb{E}[\mathbf{I}]$ as the following,

$$
\begin{aligned}
& \mathbb{E}\left\langle \widehat{\nabla}_{\mathbf{y}}f(\mathbf{x}^{(t+1)}, \mathbf{y}^{(t)}) - \widehat{\nabla}_{\mathbf{y}}f(\mathbf{x}^{(t)}, \mathbf{y}^{(t)}), \mathbf{y}^{(t+1)} - \mathbf{y}^{(t)}\right\rangle \\
\leq & \frac{3}{2\gamma}\mathbb{E}\|\widehat{\nabla}_{\mathbf{y}}f(\mathbf{x}^{(t+1)}, \mathbf{y}^{(t)}) - \nabla_{\mathbf{y}}f_{\mu,\mathbf{y}}(\mathbf{x}^{(t+1)}, \mathbf{y}^{(t)})\|^2 + \frac{\gamma}{6}\mathbb{E}\|\mathbf{y}^{(t+1)} - \mathbf{y}^{(t)}\|^2 \\
& + \frac{3}{2\gamma}\mathbb{E}\|\nabla_{\mathbf{y}}f_{\mu,\mathbf{y}}(\mathbf{x}^{(t+1)}, \mathbf{y}^{(t)}) - \nabla_{\mathbf{y}}f_{\mu,\mathbf{y}}(\mathbf{x}^{(t)}, \mathbf{y}^{(t)})\|^2 + \frac{\gamma}{6}\mathbb{E}\|\mathbf{y}^{(t+1)} - \mathbf{y}^{(t)}\|^2 \\
& + \frac{3}{2\gamma}\mathbb{E}\|\nabla_{\mathbf{y}}f_{\mu,\mathbf{y}}(\mathbf{x}^{(t)}, \mathbf{y}^{(t)}) - \widehat{\nabla}f_{\mathbf{y}}(\mathbf{x}^{(t)}, \mathbf{y}^{(t)})\|^2 + \frac{\gamma}{6}\mathbb{E}\|\mathbf{y}^{(t+1)} - \mathbf{y}^{(t)}\|^2 \\
\leq & \frac{3\sigma_y^2}{\gamma} + \frac{3L_x^2}{2\gamma}\mathbb{E}\|\mathbf{x}^{(t+1)} - \mathbf{x}^{(t)}\|^2 + \frac{\gamma}{2}\mathbb{E}\|\mathbf{y}^{(t+1)} - \mathbf{y}^{(t)}\|^2
\end{aligned}
\tag{72}
$$

where Lemma 1 is used.

Second, we need to give an upper bound of $\mathbb{E}[\mathbf{II}]$ as follows:

$$\langle \widehat{\nabla} f(\mathbf{x}^{(t)}, \mathbf{y}^{(t)}) - \widehat{\nabla} f(\mathbf{x}^{(t)}, \mathbf{y}^{(t-1)}), \mathbf{y}^{(t+1)} - \mathbf{y}^{(t)} \rangle$$

$$= \langle \widehat{\nabla} f(\mathbf{x}^{(t)}, \mathbf{y}^{(t)}) - \widehat{\nabla} f(\mathbf{x}^{(t)}, \mathbf{y}^{(t-1)}), \mathbf{v}^{(t+1)} + \mathbf{y}^{(t)} - \mathbf{y}^{(t-1)} \rangle$$

$$= \left\langle \nabla f(\mathbf{x}^{(t)}, \mathbf{y}^{(t)}) - \nabla f(\mathbf{x}^{(t)}, \mathbf{y}^{(t-1)}), \mathbf{y}^{(t)} - \mathbf{y}^{(t-1)} \right\rangle$$

$$+ \left\langle \nabla f_{\mu,\mathbf{y}}(\mathbf{x}^{(t)}, \mathbf{y}^{(t)}) - \nabla f(\mathbf{x}^{(t)}, \mathbf{y}^{(t)}), \mathbf{y}^{(t)} - \mathbf{y}^{(t-1)} \right\rangle$$

$$+ \left\langle \widehat{\nabla} f(\mathbf{x}^{(t)}, \mathbf{y}^{(t)}) - \nabla f_{\mu,\mathbf{y}}(\mathbf{x}^{(t)}, \mathbf{y}^{(t)}), \mathbf{y}^{(t)} - \mathbf{y}^{(t-1)} \right\rangle$$

$$- \left\langle \nabla f_{\mu,\mathbf{y}}(\mathbf{x}^{(t)}, \mathbf{y}^{(t-1)}) - \nabla f(\mathbf{x}^{(t)}, \mathbf{y}^{(t-1)}), \mathbf{y}^{(t)} - \mathbf{y}^{(t-1)} \right\rangle$$

$$- \left\langle \widehat{\nabla} f(\mathbf{x}^{(t)}, \mathbf{y}^{(t-1)}) - \nabla f_{\mu,\mathbf{y}}(\mathbf{x}^{(t)}, \mathbf{y}^{(t-1)}), \mathbf{y}^{(t)} - \mathbf{y}^{(t-1)} \right\rangle$$

$$+ \langle \widehat{\nabla} f(\mathbf{x}^{(t)}, \mathbf{y}^{(t)}) - \widehat{\nabla} f(\mathbf{x}^{(t)}, \mathbf{y}^{(t-1)}), \mathbf{v}^{(t+1)} \rangle.$$

Next, we take expectation on both sides of the above equality and obtain

$$\mathbb{E} \langle \widehat{\nabla} f(\mathbf{x}^{(t)}, \mathbf{y}^{(t)}) - \widehat{\nabla} f(\mathbf{x}^{(t)}, \mathbf{y}^{(t-1)}), \mathbf{y}^{(t+1)} - \mathbf{y}^{(t)} \rangle$$

$$\overset{(a)}{\leq} \left( \frac{3\beta L_y^2}{2} + \beta \right) \| \mathbf{y}^{(t)} - \mathbf{y}^{(t-1)} \|^2 + \frac{1}{2\beta} \| \mathbf{v}^{(t+1)} \|^2 - \gamma \| \mathbf{y}^{(t)} - \mathbf{y}^{(t-1)} \|^2$$

$$+ \frac{\mu^2 d^2 L_y^2}{4\beta} + 4\beta \sigma_y^2 \tag{73}$$

where in $(a)$ we use the fact that 1) $\gamma$-strong concavity of $f$ with respect to $\mathbf{y}$:

$$\left\langle \nabla f(\mathbf{x}^{(t)}, \mathbf{y}^{(t)}) - \nabla f(\mathbf{x}^{(t)}, \mathbf{y}^{(t-1)}), \mathbf{y}^{(t)} - \mathbf{y}^{(t-1)} \right\rangle \leq -\gamma \| \mathbf{y}^{(t)} - \mathbf{y}^{(t-1)} \|^2; \tag{74}$$

and the facts that 2) smoothing property (26) and Young's inequality

$$\mathbb{E} \left\langle \nabla f_{\mu,\mathbf{y}}(\mathbf{x}^{(t)}, \mathbf{y}^{(t)}) - \nabla f(\mathbf{x}^{(t)}, \mathbf{y}^{(t)}), \mathbf{y}^{(t)} - \mathbf{y}^{(t-1)} \right\rangle \leq \frac{\mu^2 d^2 L_y^2}{8\beta} + \frac{\beta}{2} \| \mathbf{y}^{(t)} - \mathbf{y}^{(t-1)} \|^2; \tag{75}$$

and the fact that 3) the ZO estimator is unbiased according to Lemma 1

$$\mathbb{E} \left\langle \widehat{\nabla} f(\mathbf{x}^{(t)}, \mathbf{y}^{(t)}) - \nabla f_{\mu,\mathbf{y}}(\mathbf{x}^{(t)}, \mathbf{y}^{(t)}), \mathbf{y}^{(t)} - \mathbf{y}^{(t-1)} \right\rangle = 0; \tag{76}$$

and

$$\mathbb{E} \left\langle \nabla f_{\mu,\mathbf{y}}(\mathbf{x}^{(t)}, \mathbf{y}^{(t-1)}) - \nabla f(\mathbf{x}^{(t)}, \mathbf{y}^{(t-1)}), \mathbf{y}^{(t)} - \mathbf{y}^{(t-1)} \right\rangle \leq \frac{\mu^2 d^2 L_y^2}{8\beta} + \frac{\beta}{2} \| \mathbf{y}^{(t)} - \mathbf{y}^{(t-1)} \|^2; \tag{77}$$

and from Corollary 1 we have

$$\mathbb{E} \left\langle \widehat{\nabla} f(\mathbf{x}^{(t)}, \mathbf{y}^{(t-1)}) - \nabla f_{\mu,\mathbf{y}}(\mathbf{x}^{(t)}, \mathbf{y}^{(t-1)}), \mathbf{y}^{(t)} - \mathbf{y}^{(t-1)} \right\rangle \leq \beta \sigma_y^2; \tag{78}$$

and

$$\mathbb{E} \langle \widehat{\nabla} f(\mathbf{x}^{(t)}, \mathbf{y}^{(t)}) - \widehat{\nabla} f(\mathbf{x}^{(t)}, \mathbf{y}^{(t-1)}), \mathbf{v}^{(t+1)} \rangle$$

$$\leq \frac{3\beta}{2} \mathbb{E} \| \nabla f_{\mu,\mathbf{y}}(\mathbf{x}^{(t)}, \mathbf{y}^{(t)}) - \widehat{\nabla} f(\mathbf{x}^{(t)}, \mathbf{y}^{(t)}) \|^2 + \frac{1}{6\beta} \| \mathbf{v}^{(t+1)} \|^2$$

$$+ \frac{3\beta}{2} \mathbb{E} \| \nabla f_{\mu,\mathbf{y}}(\mathbf{x}^{(t)}, \mathbf{y}^{(t)}) - \nabla f_{\mu,\mathbf{y}}(\mathbf{x}^{(t)}, \mathbf{y}^{(t-1)}) \|^2 + \frac{1}{6\beta} \| \mathbf{v}^{(t+1)} \|^2$$

$$+ \frac{3\beta}{2} \mathbb{E} \| \nabla f_{\mu,\mathbf{y}}(\mathbf{x}^{(t)}, \mathbf{y}^{(t-1)}) - \widehat{\nabla} f(\mathbf{x}^{(t)}, \mathbf{y}^{(t-1)}) \|^2 + \frac{1}{6\beta} \| \mathbf{v}^{(t+1)} \|^2$$

$$\leq 3\beta \sigma_y^2 + \frac{1}{2\beta} \| \mathbf{v}^{(t+1)} \|^2 + \frac{3\beta L_y^2}{2} \| \mathbf{y}^{(t)} - \mathbf{y}^{(t-1)} \|^2. \tag{79}$$

Then, from (71), we can have

$$
\begin{aligned}
\frac{1}{2\beta}\mathbb{E}\|\mathbf{y}^{(t+1)} - \mathbf{y}^{(t)}\|^2 \leq & \frac{1}{2\beta}\mathbb{E}\|\mathbf{y}^{(t)} - \mathbf{y}^{(t-1)}\|^2 - \frac{1}{2\beta}\mathbb{E}\|\mathbf{v}^{(t+1)}\|^2 \\
& + \frac{3\sigma_y^2}{\gamma} + \frac{3L_x^2}{2\gamma}\mathbb{E}\|\mathbf{x}^{(t+1)} - \mathbf{x}^{(t)}\|^2 + \frac{\gamma}{2}\mathbb{E}\|\mathbf{y}^{(t+1)} - \mathbf{y}^{(t)}\|^2 \\
& + \left\langle \widehat{\nabla} f(\mathbf{x}^{(t)}, \mathbf{y}^{(t)}) - \widehat{\nabla} f(\mathbf{x}^{(t)}, \mathbf{y}^{(t-1)}), \mathbf{y}^{(t+1)} - \mathbf{y}^{(t)} \right\rangle \\
\leq & \frac{1}{2\beta}\mathbb{E}\|\mathbf{y}^{(t)} - \mathbf{y}^{(t-1)}\|^2 + \frac{\gamma}{2}\mathbb{E}\|\mathbf{y}^{(t+1)} - \mathbf{y}^{(t)}\|^2 \\
& + \frac{3L_y^2}{2\gamma}\mathbb{E}\|\mathbf{x}^{(t+1)} - \mathbf{x}^{(t)}\|^2 - \left(\gamma - \left(\frac{3\beta L_y^2}{2} + \beta\right)\right)\mathbb{E}\|\mathbf{y}^{(t)} - \mathbf{y}^{(t-1)}\|^2 \\
& + \frac{3\sigma_y^2}{\gamma} + 4\beta\sigma_y^2 + \frac{\mu^2 d^2 L_y^2}{4\beta},
\end{aligned}
\tag{80}
$$

which implies

$$
\begin{aligned}
\frac{2}{\beta^2\gamma}\mathbb{E}\|\mathbf{y}^{(t+1)} - \mathbf{y}^{(t)}\|^2 \leq & \frac{2}{\beta^2\gamma}\mathbb{E}\|\mathbf{y}^{(t)} - \mathbf{y}^{(t-1)}\|^2 + \frac{2}{\beta}\mathbb{E}\|\mathbf{y}^{(t+1)} - \mathbf{y}^{(t)}\|^2 \\
& + \frac{6L_y^2}{\beta\gamma^2}\mathbb{E}\|\mathbf{x}^{(t+1)} - \mathbf{x}^{(t)}\|^2 - \left(\frac{4}{\beta} - \frac{6L_y^2 + 4}{\gamma}\right)\mathbb{E}\|\mathbf{y}^{(t)} - \mathbf{y}^{(t-1)}\|^2 \\
& + \frac{4\sigma_y^2}{\beta\gamma}\left(\frac{3}{\gamma} + 4\beta\right) + \frac{\mu^2 d^2 L_y^2}{\beta^2\gamma}.
\end{aligned}
\tag{81}
$$

### A.4.1   Proof of Lemma 4

**Proof**: Similarly as A.3.2, let $f'(\mathbf{x}^{(t+1)}, \mathbf{y}^{(t+1)}) = f(\mathbf{x}^{(t+1)}, \mathbf{y}^{(t+1)}) - 1(\mathbf{y}^{(t+1)})$, $1(\cdot)$ denotes the indicator function and $\xi^{(t)}$ denote the subgradient of $1(\mathbf{y}^{(t)})$. Since function $f'(\mathbf{x}, \mathbf{y})$ is concave with respect to $\mathbf{y}$, we have

$$
\begin{aligned}
f'(\mathbf{x}^{(t+1)}, \mathbf{y}^{(t+1)}) - f'(\mathbf{x}^{(t+1)}, \mathbf{y}^{(t)}) & \leq \langle \nabla f(\mathbf{x}^{(t+1)}, \mathbf{y}^{(t)}), \mathbf{y}^{(t+1)} - \mathbf{y}^{(t)} \rangle - \langle \xi^{(t)}, \mathbf{y}^{(t+1)} - \mathbf{y}^{(t)} \rangle \\
& \overset{(a)}{=} \frac{1}{\beta}\|\mathbf{y}^{(t+1)} - \mathbf{y}^{(t)}\|^2 - \langle \xi^{(t)} - \xi^{(t+1)}, \mathbf{y}^{(t+1)} - \mathbf{y}^{(t)} \rangle \\
& = \frac{1}{\beta}\|\mathbf{y}^{(t+1)} - \mathbf{y}^{(t)}\|^2 + \left\langle \widehat{\nabla} f(\mathbf{x}^{(t+1)}, \mathbf{y}^{(t)}) - \widehat{\nabla} f(\mathbf{x}^{(t)}, \mathbf{y}^{(t-1)}), \mathbf{y}^{(t+1)} - \mathbf{y}^{(t)} \right\rangle \\
& \quad - \frac{1}{\beta}\left\langle \mathbf{v}^{(t+1)}, \mathbf{y}^{(t+1)} - \mathbf{y}^{(t)} \right\rangle
\end{aligned}
\tag{82}
$$

where in $(a)$ we use $\xi^{(t+1)} = \widehat{\nabla} f(\mathbf{x}^{(t+1)}, \mathbf{y}^{(t)}) - \frac{1}{\beta}(\mathbf{y}^{(t+1)} - \mathbf{y}^{(t)})$. Then, we have

$$
\begin{aligned}
& \mathbb{E}f(\mathbf{x}^{(t+1)}, \mathbf{y}^{(t+1)}) - \mathbb{E}f(\mathbf{x}^{(t+1)}, \mathbf{y}^{(t)}) + \frac{1}{\beta}\left\langle \mathbf{v}^{(t+1)}, \mathbf{y}^{(t+1)} - \mathbf{y}^{(t)} \right\rangle \\
& \leq \frac{1}{\beta}\|\mathbf{y}^{(t+1)} - \mathbf{y}^{(t)}\|^2 + \left\langle \widehat{\nabla} f(\mathbf{x}^{(t+1)}, \mathbf{y}^{(t)}) - \widehat{\nabla} f(\mathbf{x}^{(t)}, \mathbf{y}^{(t-1)}), \mathbf{y}^{(t+1)} - \mathbf{y}^{(t)} \right\rangle.
\end{aligned}
$$

Applying the steps from (73) to (80), we can have

$$
\begin{aligned}
& \mathbb{E}f(\mathbf{x}^{(t+1)}, \mathbf{y}^{(t+1)}) - \mathbb{E}f(\mathbf{x}^{(t+1)}, \mathbf{y}^{(t)}) \\
& \leq \frac{1}{\beta}\mathbb{E}\|\mathbf{y}^{(t+1)} - \mathbf{y}^{(t)}\|^2 + \frac{1}{2\beta}\mathbb{E}\|\mathbf{y}^{(t)} - \mathbf{y}^{(t-1)}\|^2 - \left(\gamma - \left(\frac{3\beta L_y^2}{2} + \beta\right)\right)\|\mathbf{y}^{(t)} - \mathbf{y}^{(t-1)}\|^2 \\
& \quad + \frac{3\beta L_x^2}{2}\mathbb{E}\|\mathbf{x}^{(t+1)} - \mathbf{x}^{(t)}\|^2 + 7\beta\sigma_y^2 + \frac{\mu^2 d^2 L_y^2}{4\beta}
\end{aligned}
\tag{83}
$$

where we use

$$\mathbb{E}\left\langle \widehat{\nabla}_{\mathbf{y}} f(\mathbf{x}^{(t+1)}, \mathbf{y}^{(t)}) - \widehat{\nabla}_{\mathbf{y}} f(\mathbf{x}^{(t)}, \mathbf{y}^{(t)}), \mathbf{y}^{(t+1)} - \mathbf{y}^{(t)} \right\rangle$$

$$\leq 3\beta\sigma_y^2 + \frac{3\beta L_x^2}{2}\mathbb{E}\|\mathbf{x}^{(t+1)} - \mathbf{x}^{(t)}\|^2 + \frac{1}{2\beta}\mathbb{E}\|\mathbf{y}^{(t+1)} - \mathbf{y}^{(t)}\|^2. \tag{84}$$

Combing (81), we have

$$\mathbb{E}f(\mathbf{x}^{(t+1)}, \mathbf{y}^{(t+1)}) + \left(\frac{2}{\beta^2\gamma} + \frac{1}{2\beta}\right)\mathbb{E}\|\mathbf{y}^{(t+1)} - \mathbf{y}^{(t)}\|^2 - \left(\frac{4}{\beta} - \frac{6L_y^2 + 4}{\gamma}\right)\mathbb{E}\|\mathbf{y}^{(t+1)} - \mathbf{y}^{(t)}\|^2$$

$$\leq \mathbb{E}f(\mathbf{x}^{(t+1)}, \mathbf{y}^{(t)}) + \left(\frac{2}{\beta^2\gamma} + \frac{1}{2\beta}\right)\mathbb{E}\|\mathbf{y}^{(t)} - \mathbf{y}^{(t-1)}\|^2 - \left(\frac{4}{\beta} - \frac{6L_y^2 + 4}{\gamma}\right)\mathbb{E}\|\mathbf{y}^{(t)} - \mathbf{y}^{(t-1)}\|^2$$

$$- \left(\frac{1}{2\beta} - \frac{6L_y^2 + 4}{\gamma}\right)\mathbb{E}\|\mathbf{y}^{(t+1)} - \mathbf{y}^{(t)}\|^2 + \left(\frac{6L_x^2}{\gamma^2\beta} + \frac{3\beta L_x^2}{2}\right)\mathbb{E}\|\mathbf{x}^{(t+1)} - \mathbf{x}^{(t)}\|^2.$$

$$+ \frac{\mu^2 d^2 L_y^2}{\beta}\left(\frac{1}{4} + \frac{1}{\beta\gamma}\right) + \left(7\beta + \frac{4}{\beta\gamma}\left(\frac{3}{\gamma} + 7\beta\right)\right)\sigma_y^2. \tag{85}$$

□

### A.4.2 PROOF OF THEOREM 2

**Proof**: From (37), we know the "descent" of the minimization step, i.e., the changes from $\mathcal{P}'(\mathbf{x}^{(t)}, \mathbf{y}^{(t)}, \Delta_{\mathbf{y}}^{(t)})$ to $\mathcal{P}'(\mathbf{x}^{(t+1)}, \mathbf{y}^{(t)}, \Delta_{\mathbf{y}}^{(t)})$. Combining the "descent" of the maximization step by Lemma 4 shown in (85), we can obtain the following:

$$\mathcal{P}'(\mathbf{x}^{(t+1)}, \mathbf{y}^{(t+1)}, \Delta_{\mathbf{y}}^{(t+1)})$$

$$\leq \mathcal{P}'(\mathbf{x}^{(t)}, \mathbf{y}^{(t)}, \Delta_{\mathbf{y}}^{(t)}) - \underbrace{\left(\frac{1}{2\beta} - \frac{6L_y^2 + 4}{\gamma}\right)}_{a_1}\mathbb{E}\left[\|\mathbf{y}^{(t+1)} - \mathbf{y}^{(t)}\|^2\right] \tag{86}$$

$$- \underbrace{\left(\frac{1}{\alpha} - \left(\frac{L_x}{2} + \frac{6L_x^2}{\gamma^2\beta} + \frac{3\beta L_x^2}{2}\right)\right)}_{a_2}\mathbb{E}\left[\|\mathbf{x}^{(t+1)} - \mathbf{x}^{(t)}\|^2\right]$$

$$+ \mu^2\underbrace{\left(L_x + \frac{d^2 L_y^2}{\beta}\left(\frac{1}{4} + \frac{1}{\beta\gamma}\right)\right)}_{b_1} + \alpha\sigma_x^2 + \underbrace{\left(7\beta + \frac{4}{\beta\gamma}\left(\frac{3}{\gamma} + 4\beta\right)\right)}_{b_2}\sigma_y^2.$$

When $\beta, \alpha$ satisfy the following conditions:

$$\beta < \frac{\gamma}{4(3L_y^2 + 2)}, \quad \text{and} \quad \alpha < \frac{1}{\frac{L_x}{2} + \frac{6L_x^2}{\gamma^2\beta} + \frac{3\beta L_x^2}{2}}, \tag{87}$$

we can conclude that there exist $b_1, b_2 > 0$ such that

$$\mathcal{P}'(\mathbf{x}^{(t+1)}, \mathbf{y}^{(t+1)}, \Delta_{\mathbf{y}}^{(t+1)})$$

$$\leq \mathcal{P}'(\mathbf{x}^{(t)}, \mathbf{y}^{(t)}, \Delta_{\mathbf{y}}^{(t)}) - a_1\mathbb{E}\left[\|\mathbf{y}^{(t+1)} - \mathbf{y}^{(t)}\|^2\right]$$

$$- a_2\left[\|\mathbf{x}^{(t+1)} - \mathbf{x}^{(t)}\|^2\right] + b_1\mu^2 + \alpha\sigma_x^2 + b_2\sigma_y^2$$

$$\leq -\zeta'\mathbb{E}\left[\|\mathbf{y}^{(t+1)} - \mathbf{y}^{(t)}\|^2 + \|\mathbf{x}^{(t+1)} - \mathbf{x}^{(t)}\|^2\right] + b_1\mu^2 + \alpha\sigma_x^2 + b_2\sigma^2 \tag{88}$$

where $\zeta' = \min\{a_1, a_2\}$.

From (6), we can have

$$\mathbb{E}\|\mathcal{G}(\mathbf{x}^{(t)}, \mathbf{y}^{(t)})\|$$

$$\leq \frac{1}{\alpha}\mathbb{E}\|\mathbf{x}^{(t+1)} - \mathbf{x}^{(t)}\| + \frac{1}{\alpha}\mathbb{E}\|\mathbf{x}^{(t+1)} - \text{proj}_{\mathcal{X}}(\mathbf{x}^{(t)} - \alpha\nabla_{\mathbf{x}}f(\mathbf{x}^{(t)}, \mathbf{y}^{(t)}))\|$$

$$+ \frac{1}{\beta}\mathbb{E}\|\mathbf{y}^{(t+1)} - \mathbf{y}^{(t)}\| + \frac{1}{\beta}\mathbb{E}\|\mathbf{y}^{(t+1)} - \text{proj}_{\mathcal{Y}}(\mathbf{y}^{(t)} + \beta\nabla_{\mathbf{y}}f(\mathbf{x}^{(t)}, \mathbf{y}^{(t)}))\|$$

$$\overset{(a)}{\leq} \frac{1}{\alpha}\mathbb{E}\|\mathbf{x}^{(t+1)} - \mathbf{x}^{(t)}\| + \frac{1}{\beta}\mathbb{E}\|\mathbf{y}^{(t+1)} - \mathbf{y}^{(t)}\|$$

$$+ \frac{1}{\alpha}\mathbb{E}\|\text{proj}_{\mathcal{X}}(\mathbf{x}^{(t+1)} - \alpha(\widehat{\nabla}_{\mathbf{x}}f(\mathbf{x}^{(t)}, \mathbf{y}^{(t)}) + \frac{1}{\alpha}(\mathbf{x}^{(t+1)} - \mathbf{x}^{(t)})) - \text{proj}_{\mathcal{X}}(\mathbf{x}^{(t)} - \alpha\nabla_{\mathbf{x}}f(\mathbf{x}^{(t)}, \mathbf{y}^{(t)}))\|$$

$$+ \frac{1}{\beta}\mathbb{E}\|\text{proj}_{\mathcal{Y}}(\mathbf{y}^{(t+1)} + \beta(\widehat{\nabla}_{\mathbf{y}}f(\mathbf{x}^{(t+1)}, \mathbf{y}^{(t)}) - \frac{1}{\beta}(\mathbf{y}^{(t+1)} - \mathbf{y}^{(t)})) - \text{proj}_{\mathcal{Y}}(\mathbf{y}^{(t)} + \beta\nabla_{\mathbf{y}}f(\mathbf{x}^{(t)}, \mathbf{y}^{(t)}))\|$$

$$\overset{(b)}{\leq} \frac{3}{\alpha}\mathbb{E}\|\mathbf{x}^{(t+1)} - \mathbf{x}^{(t)}\| + \mathbb{E}\|\widehat{\nabla}_{\mathbf{x}}f(\mathbf{x}^{(t)}, \mathbf{y}^{(t)})) - \nabla_{\mathbf{x}}f(\mathbf{x}^{(t)}, \mathbf{y}^{(t)}))\|$$

$$+ \frac{3}{\beta}\mathbb{E}\|\mathbf{y}^{(t+1)} - \mathbf{y}^{(t)}\| + \mathbb{E}\|\widehat{\nabla}_{\mathbf{y}}f(\mathbf{x}^{(t+1)}, \mathbf{y}^{(t)}) - \nabla_{\mathbf{y}}f(\mathbf{x}^{(t)}, \mathbf{y}^{(t)})\|$$

$$\leq \frac{3}{\alpha}\mathbb{E}\|\mathbf{x}^{(t+1)} - \mathbf{x}^{(t)}\| + \mathbb{E}\|\widehat{\nabla}_{\mathbf{x}}f(\mathbf{x}^{(t)}, \mathbf{y}^{(t)})) - \nabla_{\mathbf{x}}f_{\mu,\mathbf{y}}(\mathbf{x}^{(t)}, \mathbf{y}^{(t)}))\|$$

$$+ \mathbb{E}\|\nabla_{\mathbf{x}}f_{\mu,\mathbf{y}}(\mathbf{x}^{(t)}, \mathbf{y}^{(t)})) - \nabla_{\mathbf{x}}f(\mathbf{x}^{(t)}, \mathbf{y}^{(t)}))\|$$

$$+ \frac{3}{\beta}\mathbb{E}\|\mathbf{y}^{(t+1)} - \mathbf{y}^{(t)}\| + \mathbb{E}\|\widehat{\nabla}_{\mathbf{y}}f(\mathbf{x}^{(t+1)}, \mathbf{y}^{(t)}) - \nabla_{\mathbf{y}}f_{\mu,\mathbf{y}}(\mathbf{x}^{(t+1)}, \mathbf{y}^{(t)})\|$$

$$+ \mathbb{E}\|\nabla_{\mathbf{y}}f_{\mu,\mathbf{y}}(\mathbf{x}^{(t+1)}, \mathbf{y}^{(t)}) - \nabla_{\mathbf{y}}f_{\mu,\mathbf{y}}(\mathbf{x}^{(t)}, \mathbf{y}^{(t)})\|$$

$$+ \mathbb{E}\|\nabla_{\mathbf{y}}f_{\mu,\mathbf{y}}(\mathbf{x}^{(t)}, \mathbf{y}^{(t)}) - \nabla_{\mathbf{y}}f(\mathbf{x}^{(t)}, \mathbf{y}^{(t)})\|$$

$$\overset{(c)}{\leq} \left(\frac{3}{\alpha} + L_x\right)\mathbb{E}\|\mathbf{x}^{(t+1)} - \mathbf{x}^{(t)}\| + \frac{3}{\beta}\mathbb{E}\|\mathbf{y}^{(t+1)} - \mathbf{y}^{(t)}\| + 2\sigma_y^2 + \mu^2 d^2 L_y^2$$

where in $(a)$ we use the optimality condition of $\mathbf{x}^{(t)}$-subproblem; in $(b)$ we use nonexpansiveness of the projection operator; in $(c)$ we apply the Lipschitz continuous of function $f(\mathbf{x}, \mathbf{y})$ under assumption **A2**.

Therefore, we can know that

$$\mathbb{E}\left[\|\mathcal{G}(\mathbf{x}^{(t)}, \mathbf{y}^{(t)})\|^2\right] \leq c\left(\|\mathbf{x}^{(t+1)} - \mathbf{x}^{(t)}\|^2 + \|\mathbf{y}^{(t+1)} - \mathbf{y}^{(t)}\|^2\right) + 2\sigma_y^2 + \mu^2 d^2 L_y^2. \tag{89}$$

After applying the telescope sum on (88) and taking expectation over (89), we have

$$\frac{1}{T}\sum_{t=1}^{T}\mathbb{E}\left[\|\mathcal{G}(\mathbf{x}^{(t)}, \mathbf{y}^{(t)})\|^2\right] \leq \frac{c}{\zeta'}\frac{\mathcal{P}_1 - \mathcal{P}_{T+1}}{T} + \frac{cb_1}{\zeta'}\mu^2 + \frac{c\alpha\sigma_x^2}{\zeta'} + \frac{cb_2}{\zeta'}\sigma_y^2 + 2\sigma_y^2 + \mu^2 d^2 L_y^2. \tag{90}$$

Recall from **A1** that $f \geq f^*$ and $\mathcal{Y}$ is bounded with diameter $R$, therefore, $\mathcal{P}_t$ given by (11) yields

$$\mathcal{P}_t \geq f^* + \left(\frac{\min\{4 + 4(3L_y^2 + 2)\beta^2 - 7\beta\gamma, 0\}}{\beta^2\gamma}\right)R^2, \quad \forall t. \tag{91}$$

And let $(\mathbf{x}^{(r)}, \mathbf{y}^{(r)})$ be uniformly and randomly picked from $\{(\mathbf{x}^{(t)}, \mathbf{y}^{(t)})\}_{t=1}^{T}$, based on (91) and (90), we obtain

$$\mathbb{E}_r\left[\mathbb{E}\left[\|\mathcal{G}(\mathbf{x}^{(r)}, \mathbf{y}^{(r)})\|^2\right]\right] = \frac{1}{T}\sum_{t=1}^{T}\mathbb{E}\left[\|\mathcal{G}(\mathbf{x}^{(t)}, \mathbf{y}^{(t)})\|^2\right]$$

$$\leq \frac{c}{\zeta'}\frac{\mathcal{P}_1 - f^* - \nu'R^2}{T} + \frac{cb_1}{\zeta'}\mu^2 + \frac{c\alpha\sigma_x^2}{\zeta'} + \frac{cb_2}{\zeta'}\sigma_y^2 + 2\sigma_y^2 + \mu^2 d^2 L_y^2 \tag{92}$$

where recall that $\zeta' = \min\{a_1, a_2\}$, $c = \max\{L_x + \frac{3}{\alpha}, \frac{3}{\beta}\}$, and $\nu' = \frac{\min\{4 + 4(3L_y^2 + 2)\beta^2 - 7\beta\gamma, 0\}}{\beta^2\gamma}$.

The proof is now complete. $\square$

## B  TOY EXAMPLE IN BOGUNOVIC ET AL. (2018): ZO-MIN-MAX VERSUS BO

We review the example in Bogunovic et al. (2018) as below,

$$
\underset{\mathbf{x}\in\mathcal{C}}{\text{maximize}}\;\underset{\|\boldsymbol{\delta}\|_2\le 0.5}{\text{minimize}}\quad f(\mathbf{x}-\boldsymbol{\delta}):= -2(x_1-\delta_1)^6 + 12.2(x_1-\delta_1)^5 - 21.2(x_1-\delta_1)^4
$$

$$
\begin{aligned}
&-6.2(x_1-\delta_1) + 6.4(x_1-\delta_1)^3 + 4.7(x_1-\delta_1)^2 - (x_2-\delta_2)^6\\
&+11(x_2-\delta_2)^5 - 43.3(x_2-\delta_2)^4 + 10(x_2-\delta_2) + 74.8(x_2-\delta_2)^3\\
&-56.9(x_2-\delta_2)^2 + 4.1(x_1-\delta_1)(x_2-\delta_2) + 0.1(x_1-\delta_1)^2(x_2-\delta_2)^2\\
&-0.4(x_2-\delta_2)^2(x_1-\delta_1) - 0.4(x_1-\delta_1)^2(x_2-\delta_2),
\end{aligned}
\tag{93}
$$

where $\mathbf{x} \in \mathbb{R}^2$, and $\mathcal{C} = \{x_1 \in (-0.95, 3.2), x_2 \in (-0.45, 4.4)\}$.

Problem (93) can be equivalently transformed to the min-max setting consistent with ours

$$
\underset{\mathbf{x}\in\mathcal{C}}{\text{minimize}}\;\underset{\|\boldsymbol{\delta}\|_2\le 0.5}{\text{maximize}}\quad -f(\mathbf{x}-\boldsymbol{\delta}).
\tag{94}
$$

The optimality of solving problem (93) is measured by regret versus iteration $t$,

$$
\text{Regret}(t) = \underset{\|\boldsymbol{\delta}\|_2\le 0.5}{\text{minimize}}\, f(\mathbf{x}^*-\boldsymbol{\delta}) - \underset{\|\boldsymbol{\delta}\|_2\le 0.5}{\text{minimize}}\, f(\mathbf{x}^{(t)}-\boldsymbol{\delta}),
\tag{95}
$$

where $\text{minimize}_{\|\boldsymbol{\delta}\|_2\le 0.5}\, f(\mathbf{x}^*-\boldsymbol{\delta}) = -4.33$ and $\mathbf{x}^* = [-0.195, 0.284]^T$ Bogunovic et al. (2018).

In Figure A1, we compare the convergence performance and computation time of ZO-Min-Max with the BO based approach STABLEOPT proposed in Bogunovic et al. (2018). Here we choose the same initial point for both ZO-Min-Max and STABLEOPT. And we set the same number of function queries per iteration for ZO-Min-Max (with $q = 1$) and STABLEOPT. We recall from (2) that the larger $q$ is, the more queries ZO-Min-Max takes. In our experiments, we present the best achieved regret up to time $t$ and report the average performance of each method over 5 random trials. As we can see, ZO-Min-Max is more stable, with lower regret and less running time. Besides, as $q$ becomes larger, ZO-Min-Max has a faster convergence rate. We remark that BO is slow since learning the accurate GP model and solving the acquisition problem takes intensive computation cost.

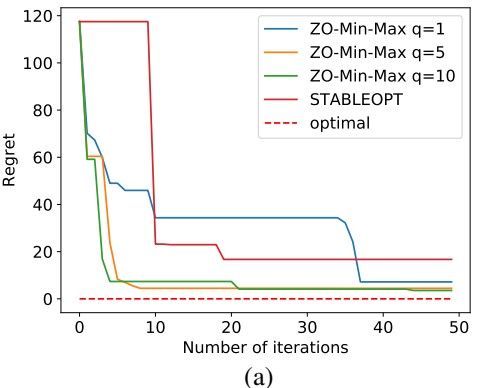
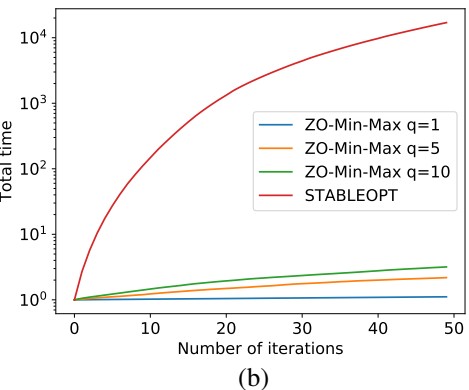

(a)  (b)

**Figure A1:** Comparison of ZO-Min-Max against STABLEOPT Bogunovic et al. (2018): a) Convergence performance; b) Computation time (seconds).

## C  EXPERIMENT SETUP ON POISONING ATTACK

In our experiment, we generate a synthetic dataset that contains $n = 1000$ samples $(\mathbf{z}_i, t_i)$. We randomly draw the feature vector $\mathbf{z}_i \in \mathbb{R}^{100}$ from $\mathcal{N}(\mathbf{0}, \mathbf{I})$, and determine $t_i = 1$ if $1/(1 + e^{-(\mathbf{z}_i^T \boldsymbol{\theta}^* + \nu_i)}) > 0.5$. Here we choose $\boldsymbol{\theta}^* = \mathbf{1}$ as the ground-truth model parameters, and $\nu_i \in \mathcal{N}(0, 10^{-3})$ as random noise. We randomly split the generated dataset into the training dataset $\mathcal{D}_{\text{tr}}$ (70%) and the testing dataset $\mathcal{D}_{\text{te}}$ (30%). We specify our learning model as the

logistic regression model for binary classification. Thus, the loss function in problem (15) is chosen as $F_{\text{tr}}(\mathbf{x}, \boldsymbol{\theta}; \mathcal{D}_{\text{tr}}) := h(\mathbf{x}, \boldsymbol{\theta}; \mathcal{D}_{\text{tr},1}) + h(\mathbf{0}, \boldsymbol{\theta}; D_{\text{tr},2})$, where $\mathcal{D}_{\text{tr}} = \mathcal{D}_{\text{tr},1} \cup \mathcal{D}_{\text{tr},2}$, $\mathcal{D}_{\text{tr},1}$ represents the subset of the training dataset that will be poisoned, $|\mathcal{D}_{\text{tr},1}|/|\mathcal{D}_{\text{tr}}|$ denotes the poisoning ratio, $h(\mathbf{x}, \boldsymbol{\theta}; \mathcal{D}) = -(1/|\mathcal{D}|) \sum_{(\mathbf{z}_i, t_i) \in \mathcal{D}} [t_i \log(h(\mathbf{x}, \boldsymbol{\theta}; \mathbf{z}_i)) + (1 - t_i) \log(1 - h(\mathbf{x}, \boldsymbol{\theta}; \mathbf{z}_i))]$, and $h(\mathbf{x}, \boldsymbol{\theta}; \mathbf{z}_i) = 1/(1 + e^{-(\mathbf{a}_i + \mathbf{x})^T \boldsymbol{\theta}})$. In problem (15), we also set $\epsilon = 2$ and $\lambda = 10^{-3}$. In Algorithm 1, unless specified otherwise we choose the the mini-batch size $b = 100$, the number of random direction vectors $q = 5$, the learning rate $\alpha = 0.02$ and $\beta = 0.05$, and the total number of iterations $T = 50000$. We report the empirical results over 10 independent trials with random initialization.

## D    ADDITIONAL EXPERIMENT RESULTS

In Figure A2, we show how the importance weights $\mathbf{w}$ of individual attack losses are learnt during ZO-Min-Max (vs. FO-Min-Max). We can see that ZO-Min-Max takes into account different robustness levels of model-class pairs through the design of $\mathbf{w}$.

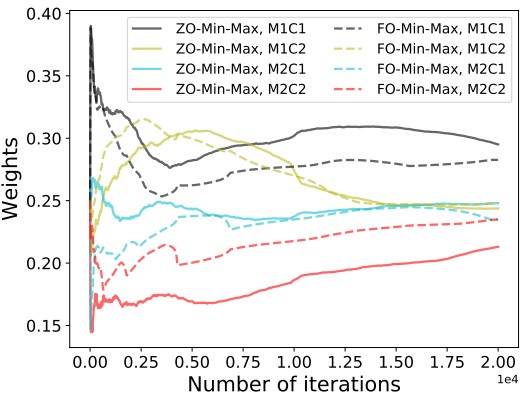

**Figure A2:** Convergence of importance weights learnt from ZO-Min-Max vs. FO-Min-Max.

In Figure A3, we contrast the success or failure (marked by blue or red in the plot) of attacking each image using the obtained universal perturbation $\mathbf{x}$ with the attacking difficulty (in terms of required iterations for successful adversarial example) of using per-image non-universal PGD attack (Madry et al., 2017b). We observe that the success rate of the ensemble universal attack is around $80\%$ at each model-class pair, where the failed cases (red cross markers) also need a large amount of iterations to succeed at the case of per-image PGD attack. And images that are difficult to attack keep consistent across models; see dash lines to associate the same images between two models in Figure A3.

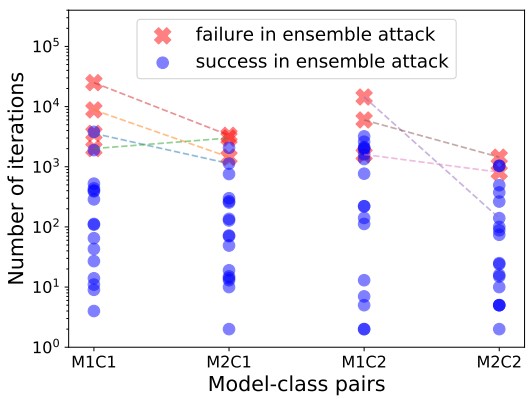

**Figure A3:** Success or failure of our ensemble attack versus successful per-image PGD attack.

In Figure A4, we show the testing accuracy of the poisoned model as the regularization parameter $\lambda$ varies. We observe that the poisoned model accuracy could be improved as $\lambda$ increases, e.g., $\lambda = 1$.

However, this leads to a decrease in clean model accuracy (below $90\%$ at $\lambda = 1$). This implies a robustness-accuracy tradeoff. If $\lambda$ continues to increase, both the clean and poisoned accuracy will decrease dramatically as the training loss in (15) is less optimized.

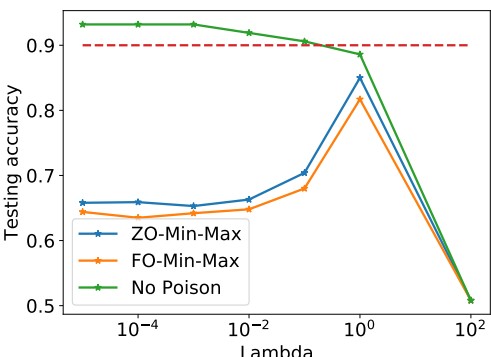

**Figure A4:** Empirical performance of ZO-Min-Max in design of poisoning attack: Testing accuracy versus regularization parameter $\lambda$.

