# OpenReview forum: "Min-Max Optimization without Gradients: Convergence and Applications to Adversarial ML"
_ICLR.cc/2020/Conference — Reject_

### Official Review · AnonReviewer1 · 2019-10-26
**Official Blind Review #1**

**Rating:** 3

**Review:**

The authors aim to propose new algorithms for min-max optimization problem when the gradients are not available and establish sublinear convergence of the algorithm. I don't think this paper can be accepted for ICLR for the following reasons:

1. For Setting (a) (One-sided black-box), the theory can be established by the same analysis for ZO optimization by optimizing y. Even by a proximal step for y, the analysis is essentially the same as ZO where an estimation of the gradient for x is conducted.

2. The assumptions A1 and A2 are hardly satisfied in ML applications, where the objective is essentially smooth. The authors should at least analyze the case where a sub/super-gradients is available.

3. Also, for most ML problems we have today, I don't find many applications where the gradients are not available, and I thus feel that it is not interesting to consider ZO optimizations.

**Experience Assessment:**

I have read many papers in this area.

**Review Assessment: Checking Correctness Of Derivations And Theory:**

I assessed the sensibility of the derivations and theory.

**Review Assessment: Checking Correctness Of Experiments:**

I assessed the sensibility of the experiments.

**Review Assessment: Thoroughness In Paper Reading:**

I read the paper at least twice and used my best judgement in assessing the paper.

---

> ### Author Response · Authors · 2019-11-11
> **Response to Reviewer #1:**
>
> Many thanks for the constructive comments. We answered your questions as below.
>
> # Question: For Setting (a) (One-sided black-box), the theory can be established by the same analysis for ZO optimization by optimizing y. Even by a proximal step for y, the analysis is essentially the same as ZO where an estimation of the gradient for x is conducted. #
>
> Response: In the revision, we have further clarified the technique challenges in convergence analysis of ZO-Min-Max. We provide more details as below.
>
> First, we would like to clarify that even at the one-sided black-box setting, ZO-Min-Max conducts alternating optimization using one-step ZO-PGD and PGA with respect to $x$ and $y$ respectively. This is different from a reduced ZO optimization problem with respect to x, namely,  problem $\min_{x} h(x) \triangleq \min_y f(x, y)$, which requires the algorithm to access $h(x)$ for a ZO gradient estimation, namely, obtaining the solution to problem $\min_y f(x, y)$ given $x$. However, it is usually non-trivial or computationally intensive to solve $\min_y f(x, y)$ directly. Thus, ZO-Min-Max is lighter in computation and is much simpler in implementation.
>
> Even by a proximal step for $y $, our analysis is also NOT the same as ZO where an estimation of the gradient for $x$ is conducted. Suppose that one manages to consider the min-max objective as a function like $h^{\prime}(x) = f(x, g(x, y^{t-1}))$, where $ g(x, y^{t-1})$ is the proximal step, in the conventional ZO analysis, the query of $h^{\prime}$ over $x = \tilde x$ for gradient estimation relies on querying $ g(x, y^{t-1})$ at $x = \tilde x$. This is different from our algorithm, where an estimation of the gradient for $x$ is conducted by fixing $y$ as a constant, leading to simpler gradient estimation but more involved convergence analysis, which is elaborated on in the following.
>
> The key difficulty stems from the alternating algorithmic nature (namely, primal-dual framework) as the problem is in the min-max form, which leads to  opposite optimization directions (minimization vs maximization) over variable x and y respectively.  Even applying ZO optimization to one side (say x), it needs to quantify the effect of ZO gradient estimation on the descent over both x and y; see the detailed technical reasons:
>
> Gradient estimate bias mitigation: classic ZO analysis is to quantify the gradient estimate $\nabla f(x^t)$ at iteration t in the sense that iterates simply move from $x^t$ to $x^{t+1}$. However, in min-max optimization, note that both $\nabla_x f(x^{t+1},y^t)$ and $\nabla_y f(x^{t+1},y^{t+1})$ are evaluated at x and y, thus, the trajectory of the proposed algorithm from t to t+1, i.e., $(x^t,y^t)->(x^{t+1},y^t)->(x^{t+1},y^{t+1})$, is different from the classic ZO minimization. If we directly applied the ZO convergence analysis, it becomes quite difficult to quantify the change of the objective value caused by optimizing y since the iterates have an intermediate state which is (x^{t+1},y^t), making the convergence analysis of min-max optimization different from the standard ZO minimization analysis.
>
> Potential function construction: Since optimizing y will increase the objective value, in this work, a new potential/Lyapunov-like function is constructed to show that the ascent of optimizing y is dominated by the descent of performing minimization over x based on this new function, where the stepsizes of both sides must be chosen properly.
>
> Therefore, the convergence analysis of ZO-Min-Max for solving one-sided black-box problem cannot be simply established using the same analysis of ZO optimization after optimizing y.
> Also, our one-sided black-box analysis is served as the preliminary results for our more general analysis in the two-sided black-box case, where the ZO gradient estimation errors in both minimization and maximization affects the convergence rate

---

> > ### Author Response · Authors · 2019-11-11
> > **Response to Reviewer #1 (continued):**
> >
> > # Question: The assumptions A1 and A2 are hardly satisfied in ML applications, where the objective is essentially smooth. The authors should at least analyze the case where a sub/super-gradients is available. #
> >
> > Response: We admit that A1 and A2 does not apply to all ML/DL problems. However, they are not restricted assumptions to the convergence analysis of ZO optimization. To the best of our knowledge, even for convergence analysis of  first-order methods to solve constrained min-max problems with nonconvex outer minimization and strongly concave inner maximization, the assumption of smoothness (namely, Lipschitz continuous gradient) is required since the changing speed of the gradients need to be bounded so that the descent of the algorithm can be quantified [Lu et al’19, Nouiehed et al’19, Rafique et al’19]. Besides min-max optimization, the smoothness assumption is needed even for analyzing the convergence of the state-of-the-art ML/DL optimizers, e.g., Adam and AdaGrad, to solve problems with only single objectives  [Chen et al, Ward et al., Staib et al]. In our experiments, the smooth assumption holds for the application of designing black-box poisoning attack on logistic regression. And the assumption on strongly concave inner maximization holds for both applications presented in this paper. Although it needs a long-term effort to bridge the gap between the theoretical analysis of ML/DL optimizers and ML/DL applications, our theoretical contributions under A1 & A2 should not be neglected.
> > We also want to highlight our contributions on applying ZO-Min-Max to solving black-box prediction-evision and poisoning attack generation problems in adversarial ML. We show that the proposed method has advantages in scalability and convergence accuracy. Specifically, in the example on generating black-box adversarial attacks against deep neural networks, ZO-Min-Max, to the best of our knowledge, is the first scalable black-box optimizer to handle the problem of size $157323 $ (corresponding to the number of pixels of ImageNet image).  In the other example of designing black-box poisoning attack on logistic regression, our proposed approach significantly outperforms BO-based baseline method STABLEOPT (Bogunovic et al., 2018).
> >
> > In the revision, we have further clarified the rationale behind our assumptions.
> >
> >
> > S. Lu, I. Tsaknakis, M. Hong, and Y. Chen. Hybrid block successive approximation for one-sidednon-convex min-max problems: Algorithms and applications.arXiv preprint arXiv:1902.08294,2019.
> > M. Nouiehed, M. Sanjabi, J. D. Lee, and M. Razaviyayn. Solving a class of non-convex min-max games using iterative first order methods.arXiv preprint arXiv:1902.08297, 2019.
> > H. Rafique, M. Liu, Q. Lin, and T. Yang. Non-convex min-max optimization: Provable algorithms and applications in machine learning.arXiv preprint arXiv:1810.02060, 2018
> > Ward, Rachel, Xiaoxia Wu, and Leon Bottou. "AdaGrad stepsizes: sharp convergence over nonconvex landscapes." International Conference on Machine Learning. (2019).
> > Chen, Xiangyi, Sijia Liu, Ruoyu Sun, and Mingyi Hong. "On the convergence of a class of adam-type algorithms for non-convex optimization." International Conference on Learning Representations (2019).
> > Staib, Matthew, Sashank J. Reddi, Satyen Kale, Sanjiv Kumar, and Suvrit Sra. "Escaping saddle points with adaptive gradient methods." International Conference on Machine Learning. (2019).
> > Ilija Bogunovic,  Jonathan Scarlett,  Stefanie Jegelka,  and Volkan Cevher.   Adversarially robust optimization with gaussian processes.  In Proc. of Advances in Neural Information Processing Systems, 2018.

---

> > > ### Author Response · Authors · 2019-11-11
> > > **Response to Reviewer #1 (continued):**
> > >
> > > # Question: Also, for most ML problems we have today, I don't find many applications where the gradients are not available, and I thus feel that it is not interesting to consider ZO optimizations.#
> > >
> > > Response: We are sorry to learn that 'it is not interesting to consider ZO optimizations.' As a matter of fact, ZO optimization has increasingly embraced for solving many machine learning problems where explicit expressions of the gradients are difficult or infeasible to obtain. We have revised our introduction to cover more applications to motivate ZO optimization. Details are provided below.
> > >
> > > First, ZO optimization serves as a powerful and practical tool for generation of prediction-evasive, black-box adversarial examples [Ilyas, Andrew, et al; Chen, Pin-Yu, et al]. Besides generating adversarial examples, this paper also shows that ZO optimization can be used to mount a successful training-phase (poisoning) in a fully black-box setting, where the adversary has no information about the ML/DL model and has to rely solely on the feedback stemming from the model input-output queries. ZO optimization can also help to solve automated ML problems, where the gradients with respect to ML pipeline configuration parameters are intractable [Aggarwal et al.].
> > >
> > > In addition to the case where the gradients are not available, ZO optimization provides computationally-efficient alternatives for first-order and second-order optimization encountered in many recent complex machine learning tasks, e.g., robust training by leveraging input gradient or curvature regularization [Finlay et al., Moosavi-Dezfooli, et al.], meta-learning [Fallah, et al.], network control and management [chen et al.], and  data processing in high dimension [Liu et al.]. Other recent applications include generating model-agnostic contrastive explanations [Dhurandhar et al.] and escaping saddle points [Flokas et al.]. They all found the powerfulness of ZO optimization.
> > >
> > > We hope that the aforementioned recent applications can better motivate the importance of ZO optimization for solving current ML problems.
> > >
> > >
> > > Ilyas, Andrew, et al. "Black-box adversarial attacks with limited queries and information." ICML18
> > > Chen, Pin-Yu, et al. "Zoo: Zeroth order optimization based black-box attacks to deep neural networks without training substitute models." Proceedings of the 10th ACM Workshop on Artificial Intelligence and Security. ACM, 2017.
> > > Aggarwal, Charu, et al. "How can AI Automate End-to-End Data Science?." arXiv preprint arXiv:1910.14436 (2019).
> > > Finlay, Chris, and Adam M. Oberman. "Scalable input gradient regularization for adversarial robustness." arXiv preprint arXiv:1905.11468 (2019).
> > > Moosavi-Dezfooli, Seyed-Mohsen, et al. "Robustness via curvature regularization, and vice versa." Proceedings of the IEEE Conference on Computer Vision and Pattern Recognition. 2019.
> > > Fallah, Alireza, Aryan Mokhtari, and Asuman Ozdaglar. "On the Convergence Theory of Gradient-Based Model-Agnostic Meta-Learning Algorithms." arXiv preprint arXiv:1908.10400(2019).
> > > Chen, Tianyi, and Georgios B. Giannakis. "Bandit convex optimization for scalable and dynamic IoT management." IEEE Internet of Things Journal 6.1 (2018): 1276-1286.
> > > Liu, Sijia, et al. "Zeroth-order online alternating direction method of multipliers: Convergence analysis and applications." arXiv preprint arXiv:1710.07804 (2017).
> > > Dhurandhar, Amit, et al. "Model Agnostic Contrastive Explanations for Structured Data." arXiv preprint arXiv:1906.00117 (2019).
> > > Flokas, Lampros, Emmanouil-Vasileios Vlatakis-Gkaragkounis, and Georgios Piliouras. "Efficiently avoiding saddle points with zero order methods: No gradients required." arXiv preprint arXiv:1910.13021 (2019).

---

### Official Review · AnonReviewer2 · 2019-10-27
**Official Blind Review #2**

**Rating:** 6

**Review:**

The paper presents an algorithm for performing min-max optimisation without gradients and analyses its convergence. The algorithm is evaluated for the min-max problems that arise in the context of adversarial attacks. The presented algorithm is a natural application of a zeroth-order gradient estimator and the authors also prove that the algorithm has a sublinear convergence rate (in a specific sense).

Considering that the algorithm merely applies the zeroth-order gradient estimator to min-max problems, the algorithm itself only makes up a somewhat novel contribution. However, to the best of my knowledge, it has not been used in this context before and personally I find the algorithm quite appealing. In fact, due to its simplicity it is essentially something that anyone could implement from scratch.

Perhaps a more important contribution is that the authors provide a fairly extensive convergence analysis, which is an important tool in analysing the algorithm and its properties. Unfortunately, it is not trivial to understand the presented convergence results and their practical implications (if any). For instance, equation (10), which is arguably one of the key equations in the paper, contains variables zeta, nu and P1, all of which depend on a number of other variables in a fairly complicated manner. The expression in (10) also contains terms that do not depend on T and it is not obvious how large these terms might be in practice (in the event that the assumptions are at least approximately true in a local region). Even though I am somewhat sceptical to the practical relevance of this convergence analysis, I recognise that it is an interesting and fascinating achievement that the authors have managed to provide a convergence analysis of an algorithm which is based on black-box min-max optimisation.


**Experience Assessment:**

I have read many papers in this area.

**Review Assessment: Checking Correctness Of Derivations And Theory:**

I assessed the sensibility of the derivations and theory.

**Review Assessment: Checking Correctness Of Experiments:**

I did not assess the experiments.

**Review Assessment: Thoroughness In Paper Reading:**

I read the paper at least twice and used my best judgement in assessing the paper.

---

> ### Author Response · Authors · 2019-11-11
> **Response to Reviewer #2:**
>
> Many thanks for the positive comments about our work. We addressed your questions as below.
>
> # Question:  Complicated parameters and practical relevance of this convergence analysis. #
>
> Response: In the revised version, we made a table in Appendix A.1 to clarify  the parameters involved in our convergence rate, and have provided the practical relevance of this convergence analysis. We summarize our main points as below.
>
>     1) Parameter $\zeta$: Since $\zeta$ appears in the denominator of the derived convergence error, it is necessary to show a non-trivial lower bound on $\zeta$. Remark 1 after Theorem 1 made a clarification on it: Such a lower bound exists under appropriate choices of the learning rates $\alpha$ and $\beta$.
>     2) Parameter $c$: Since $c$ is inversely proportional to learning rates $\alpha$ and $\beta$, to  guarantee the constant effect of the ratio $c/\xi$, it is better not to set these learning rate too small; see a specification of learning rates in Remark 1&2.
>     3) Parameter $\nu$: Since $\nu$ is non-negative and appears in terms of $-\nu R^2$, it will not make convergence rate worse.
>     4) Parameter $P_1$:  $P_1$ is the initial value of the potential function in (8). By setting the learning rate $\beta$ as Remark 2, P1 is then upper bounded by a constant determined by the initial value of the objective function, the distance of the first two updates, Lipschitz constant $L_y$ and strongly concave parameter $\gamma$.
>
> Local region bias and practical relevance: Remark 3 showed that the local region bias (namely, stationary gap)  is controlled by the mini-batch size b and the number of random direction vectors q. Thus, a large b or q can improve the iteration complexity of ZO-Min-Max, but would require $O(bq)$ times more function queries per iteration from Eq. (2). This shows the tradeoff between iteration complexity and function query complexity in ZO optimization.

---

### Official Review · AnonReviewer3 · 2019-11-03
**Official Blind Review #3**

**Rating:** 6

**Review:**

This paper considers zeroth-order method for min-max optimization (ZO-MIN-MAX) in two cases: one-sided black box (for outer minimization) and two-sided black box (for both inner maximization and outer minimization). Convergence analysis is carefully provided to show that ZO-MIN-MAX converges to a neighborhood of stationary points. Then, the authors empirically compare several methods on
1) adversarial attack on ImageNet with deep networks, and
2) black-box poisoning attack on logistic regression. The results show that ZO-MIN-MAX can provide satisfactory performance on these tasks.

In general a good paper with dense content, clear organization and writing. However, the experiment part does not seem truly convincing.
1.	What is the relationship between Eqn.(13) and the proposed ZO-MIN-MAX? It seem that in the experiment you compare using this loss ( Eqn.(13) ) against finite-sum loss, but both with ZO-MIN-MAX algorithm? In figure 1 and 2, I don’t see a competing method. So the point here is that the loss Eqn.(13)  is better, but not the proposed algorithm? I think you should compare different optimization algorithm under same loss, e.g. something like Eqn.(13)+ZO-MIN-MAX vs. Eqn.(13)+FO-MIN-MAX. This is not evident to show that ZO-MIN-MAX is better than other zero-th order methods.
2.	I would suggest comparing to more zeroth-order methods in the experiment.

From the experiments I cannot tell whether ZO-MIN-MAX is good enough compared with other methods

**Experience Assessment:**

I have read many papers in this area.

**Review Assessment: Checking Correctness Of Derivations And Theory:**

I assessed the sensibility of the derivations and theory.

**Review Assessment: Checking Correctness Of Experiments:**

I assessed the sensibility of the experiments.

**Review Assessment: Thoroughness In Paper Reading:**

I read the paper at least twice and used my best judgement in assessing the paper.

---

> ### Author Response · Authors · 2019-11-11
> **Response to Reviewer #3:**
>
> Many thanks for the positive comments about our work. We addressed your questions as below.
>
> # Question: What is the relationship between Eqn.(13) and the proposed ZO-MIN-MAX? #
>
> Response: We apologize for the confusion on Eq. (13), which is in the form of one-sided black-box optimization problem and was solved by ZO-Min-Max. The objective function of (13) is a black-box function with respect to the universal adversarial perturbation variable $x$ since the attacker in practice has no access to the configuration of a neural network model, and thus cannot perform back-propagation to obtain gradients. This is known as the balck-box attack [Ilyas, Andrew, et al; Chen, Pin-Yu, et al].
>
> Ilyas, Andrew, et al. "Black-box adversarial attacks with limited queries and information." ICML18
> Chen, Pin-Yu, et al. "Zoo: Zeroth order optimization based black-box attacks to deep neural networks without training substitute models." Proceedings of the 10th ACM Workshop on Artificial Intelligence and Security. ACM, 2017.
>
> # Question:  It seems that in the experiment you compare using this loss ( Eqn.(13) ) against finite-sum loss, but both with ZO-MIN-MAX algorithm?  In figure 1 and 2, I don’t see a competing method. So the point here is that the loss Eqn.(13)  is better, but not the proposed algorithm?  #
>
>
> Reponse: We apologize for the confusion on our first experiment to compare with the finite-sum (averaging) attack loss.
>
> First, when the finite-sum loss is applied, problem (13) reduces to the black-box optimization problem with a single objective function. It is thus not in the form of min-max optimization, and we solved it using the standard ZO gradient descent method rather than ZO-Min-Max.
>
> Second, although ZO-Min-Max and ZO-Finite-Sum correspond to different losses, they are competing methods in terms of attack performance to misclassify a neural network model. This comparison was motivated by the previous work on designing the adversarial perturbation against model ensembles  [Liu et al., 2018] in which the averaging attack loss over multiple models was considered. Thus, the proposed comparison with ZO-Finite-Sum aims to verify that the min-max formulation in (13) is reasonable ensemble attack formulation and in fact, it outperforms the conventional formulation design (using the averaging attack loss). Such a conclusion is supported by 1) the fast convergence of individual attack loss under each model-class pair (Figure 1-c) and 2) self-adjusted importance weights (in terms of w) in ensemble attack generation (Figure A2), which shows that imposing equal importance (namely, averaging) over individual attack losses might not be a good strategy for the design of ensemble attack.
>
> Moreover, Figure A3 was used to compare the success or failure of our obtained universal perturbation with the attacking difficulty of using per-image PGD attack baseline (Madry et al., 2017b). Both ZO-Finite-Sum and the  per-image PGD attack are competing methods to ZO-Finite-Sum since all of them enjoy the same purpose to misclassify a neural network model although they use different attack loss functions. Besides, to the best of our knowledge, we are not aware of any existing black-box min-max solver that is able to scale to problem (13) of size $157323 $ (corresponding to the pixel number of an ImageNet image)
>
> J. Liu, Weiming Zhang, and Nenghai Yu. Iterative ensemble adversarial attack. 2018
> Aleksander Madry, Aleksandar Makelov, Ludwig Schmidt, Dimitris Tsipras, and Adrian Vladu. Towards deep learning models resistant to adversarial attacks. arXiv preprint arXiv:1706.06083, 2017b.
>
>
>
> # Question: I think you should compare different optimization algorithm under same loss, e.g. something like Eqn.(13)+ZO-MIN-MAX vs. Eqn.(13)+FO-MIN-MAX. #
>
>
> Response: Thanks for the valuable suggestion. We have added new results in Figure 1a-1b and Figure A2 by comparing  ZO-Min-Max with FO-Min-Max.  As expected, FO-Min-Max yields faster convergence than ZO-Min-Max, however, the former has to access the full knowledge on the target neural network for computing the gradient of the attack loss. This is not a practical black-box attack setting. Moreover, it is also expected that ZO-Min-Max may converge to a neighborhood of the stationary point.

---

> > ### Author Response · Authors · 2019-11-11
> > **Response to Reviewer #3: (continued)**
> >
> > # Question: I would suggest comparing to more zeroth-order methods in the experiment. From the experiments I cannot tell whether ZO-MIN-MAX is good enough compared with other methods #
> >
> >
> > Response: Thank you for your constructive comments. We have tried our best to compare our method with existing scalable gradient-free methods for solving black-box optimization problems.
> >
> > The first example of designing black-box ensemble attack requires baseline methods to scale to problem (13) of size $157323 $. We are not aware of any existing gradient-free baseline method at this scale. For example, the off-the-shelf model-based derivative-free optimization solver COBYLA only supports problems with $2^{16} = 65536$ variables at maximum. By comparing to the suggested FO-Min-Max algorithm, we can see that the empirical convergence of ZO-Min-Max does not degrade too much. As a matter of fact, in our second example under a synthetic dataset, ZO-Min-Max significantly outperforms BO-based baseline method.
> >
> > In the second example of designing black-box poisoning attack on logistic regression, we consider a synthetic dataset with problem dimension 100. Different from the first example, the difficulty of the second one lies at the two sided black-box nature of problem (15). Since the conventional ZO solvers cannot apply to this min-max scenario, we compare ZO-Min-Max with BO-based baseline method STABLEOPT (Bogunovic et al., 2018) in Fig. 3. As we can see, ZO-Min-Max reaches a better solution with much faster convergence speed.
> >
> > We really hope that our response and the revised paper have clearly stated the advantage of ZO-Min-Max in scalability and convergence.

---

### Author Response · Authors · 2019-11-12
**Summary of our revision.**

1) In Sec. 1, we strengthened the applications of ZO optimization.
2) In Sec. 4, we motivated and clarified the technical challenges in convergence analysis of ZO-Min-Max.
3) In Sec. 5, we elaborated on our assumptions and provided clearer insights from our convergence analysis.
4) In Sec. 6, the comparison with FO-Min-Max was added and a better motivation on our baseline methods was provided in the first example. We re-organized the experiment section for clearer presentation.
5) In Appendix, Appendix A.1 and D were added to provide additional details.

---

### Decision · Program_Chairs · 2019-12-19

**Decision:**

Reject

**Comment:**

This paper proposes convergence results for zeroth-order optimization.

One of the main complaints was that ZO has limited use in ML. I appreciate the authors' response that there are cases where gradients are not easily available, especially for black-box attacks.

However, I find the limited applicability an issue for ICLR and I encourage the authors to find a conference that is more suited to that work.